# Stationary Deep Reinforcement Learning with Quantum K-spin Hamiltonian Equation

## Abstract

A foundational issue in deep reinforcement learning (DRL) is that *Bellman's optimality equation has multiple fixed points*—failing to return a consistent one. A direct evidence is the instability of existing DRL algorithms, namely, the high variance of cumulative rewards over multiple runs. As a fix of this problem, we propose a quantum K-spin Hamiltonian regularization term (H-term) to help a policy network stably find a *stationary* policy, which represents the lowest energy configuration of a system. First, we make a novel analogy between a Markov Decision Process (MDP) and a *quantum K-spin Ising model* and reformulate the objective function into a quantum K-spin Hamiltonian equation, a functional of policy that measures its energy. Then, we propose a generic actor-critic algorithm that utilizes the H-term to regularize the policy/actor network and provide Hamiltonian policy gradient calculations. Finally, on six challenging MuJoCo tasks over 20 runs, the proposed algorithm reduces the variance of cumulative rewards by $65.2\% \sim 85.6\%$ compared with those of existing algorithms.

## 1 Introduction

Deep reinforcement learning (DRL) [36] algorithms are quite unstable, namely, agents trained with different random seeds may have dramatically different performance. Existing works [1, 13, 6, 24] reported a high variance over multiple runs, thus it requires to train tens of agents and then pick the best one. Such a high variance puts a challenge on reliability and reproducibility [15, 14], limiting the wider adoption in real-world tasks.

The difficulty of guaranteeing stability lies in a foundational issue that **Bellman's optimality equation has multiple fixed points** [4, 31, 22, 18] —failing to return a consistent one. Consider MDP examples with an terminal state 0, as shown in Fig. 1 (we adapt dynamic programming examples [4, 31] into reinforcement learning settings. Observational experiments are given in Section 2.2),

- **Shortest path problem (deterministic)** in Fig. 1(a): At state 1, an agent transits to either state 1 or 0 with reward 0 or $b$, respectively. Assume the value function for state 0 is $V(0) = 0$. The Bellman's optimality equation for state 1 is $V(1) = \max\{V(1), b\}$, where any $V(1) \geq b$ is a feasible solution. If initialize $V_0(1) \geq b$, a resulting policy is that an agent at state 1 always transits back to state 1; otherwise, drives to terminal state 0 (always returns back to itself with reward 0).

- **Blackmailer's problem (stochastic)** in Fig. 1(b): At state 1, a profit maximizing blackmailer demands a cash amount $a \in (0, 1]$; a victim transits to state 1 with probability $a$ or state 0 with probability $1 - a$, respectively. At state 0, a victim always refuses to yield, i.e., $V(0) = 0$. The Bellman's optimality equation for state 1 is $V(1) = \max_a\{a + (1 - a)V(1)\}$, where any $V(1) \geq 1$ is a feasible solution. If initialize $V_0(1) > 1$, the blackmailer's policy is demanding $a \to 0$ to keep the victim at state 1; otherwise, demanding $a = 1$ that drives the victim to terminal state 0.

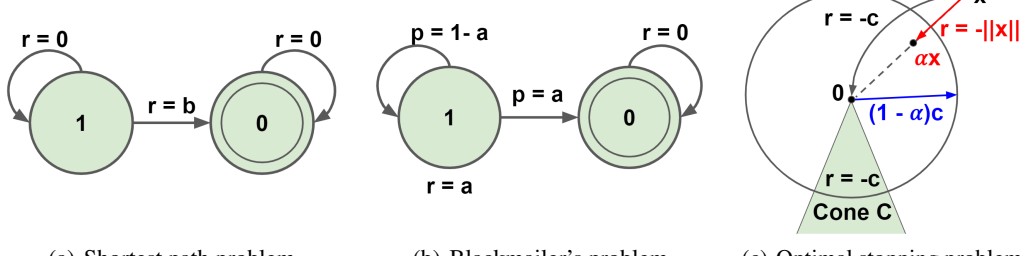

(a) Shortest path problem      (b) Blackmailer's problem      (c) Optimal stopping problem

Figure 1: MDP examples where $\gamma = 1$. Examples with $\gamma < 1$ are given in Fig. 5 in Appx. A.

- **Optimal stopping problem (terminating policies)** in Fig. 1(c): In a space $\mathbb{R}^2$ with terminal state of point 0, an agent at point $x \neq 0$ moves to either point 0 with negative reward $-c$ or point $\alpha x$ with reward $-||x||$, respectively, where $\alpha \in (0, 1)$. The Bellman's optimality equation is $V(x) = \max\{-c, -||x|| + V(\alpha x)\}$ and the optimal policy is to continue inside the sphere of radius $(1 - \alpha)c$ and to stop outside. If add a cone region $C$ within which an agent always receives a reward $-c$, a second policy is jumping to point 0 at any point in region $C$.

The instability problem has been partially addressed, such as ensemble methods [2, 8], regularization approaches [38, 9], and baseline-correction approaches [33, 42]. In particular, Generalized Advantage Estimation (GAE) [33] is a widely used one that significantly reduces the variance of the advantage function. However, they did NOT fix the foundational issue of Bellman's optimality equation in Fig. 1, thus the objective function inherently fails to search for a consistent policy. For practical usage, we often expect a DRL algorithm stably converges to a certain policy independent of initialization and noises.

As a fix of the problem, we make a novel analogy between an MDP and a *quantum K-spin Ising model* [26, 12], and reformulate a reward-maximization RL task into an energy-minimization problem, namely, finding the lowest-energy configuration of a quantum spin system. We hypothesize that *a physically stationary policy would have the lowest energy*. Different from our quantum K-spin perspective, several recent papers utilized the (classical) Hamiltonian equation to endow RL agents the capability of inductive biases. For example, [21, 40] used Hamiltonian mechanics to train an agent that learns and respects *conservation laws*; [43] applied a Hamiltonian Monte Carlo (HMC) simulator to approximate the posterior action probability; and [28] proposed an unbiased estimator for the stochastic Hamiltonian gradient methods for min-max optimization problems.

In this paper, we propose a quantum-inspired *K-spin Hamiltonian regularization term (H-term)* that helps policy gradient algorithms stably converge to a physically stationary policy. Our contributions are as follows: 1) we derive a novel regularizer, *H-term*, by a novel reformulation of the RL's objective as a K-spin Hamiltonian equation that measures the energy of a policy; 2) we propose a generic actor-critic algorithm that utilizes the H-term to regularize the policy/actor network and provide Hamiltonian policy gradient calculations on both deterministic and stochastic cases; and 3) on six challenging MuJoCo tasks over 20 runs, we show that the proposed algorithm reduces the variance of cumulative rewards by $65.2\% \sim 85.6\%$ compared with those of existing algorithms.

## 2 Bellman's Optimality Equation and Multiple Fixed Points

We describe Bellman's optimaility equation and provide observational experiments to verify the existence of multiple policies.

### 2.1 Bellman's Optimality Equation

A reinforcement learning (RL) [36] agent interacts with an unknown environment and learns an optimal policy that maximizes the cumulative reward. Mathematically, the environment can be formulated as a Markov Decision Process (MDP) with the five-tuple $\langle \mathcal{S}, \mathcal{A}, \mathbb{P}, R, \gamma \rangle$. Here $\mathcal{S}$ and $\mathcal{A}$ denote the state and action spaces; $\mathbb{P} : \mathcal{S} \times \mathcal{A} \to \Delta(\mathcal{S})$ denotes a transition probability function, where $\Delta$ is a probability simplex; $R : \mathcal{S} \times \mathcal{A} \times \mathcal{S} \to \mathbb{R}$ denotes a reward function; and $\gamma \in (0, 1]$

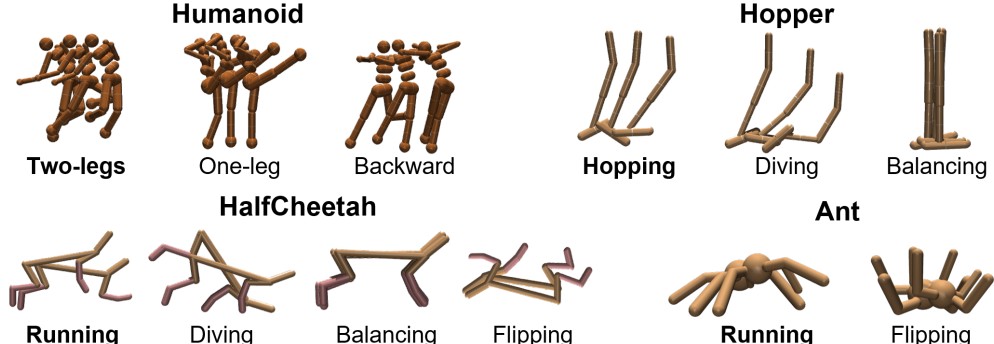

Figure 2: Different policy types for four MuJoCo [39] tasks. **Two-legs running**, one-leg running, and running backward for Humanoid; **running**, diving, and balancing for Hopper; **running**, diving, balancing and flipping for HalfCheetah; **running** and flipping for Ant. The bold ones indicate physically stationary policies.

denotes a discount factor. The objective is to find an optimal policy $\pi^* : \mathcal{S} \to \Delta(\mathcal{A})$ that maximizes (discounted) expected reward.

Consider a discrete, finite, discounted MDP with infinite horizon, one can define the Q-value function of a state-action pair $(s, a)$ under policy $\pi$ as follows

$$Q^\pi(s, a) = \mathbb{E}_{S_{k+1} \sim \mathbb{P}(\cdot | S_k, A_k), A_{k+1} \sim \pi(S_{k+1}, \cdot)} \left[ \sum_{k=0}^{\infty} \gamma^k \cdot R(S_k, A_k, S_{k+1}) | S_0 = s, A_0 = a \right], \quad (1)$$

where $R(S_k, A_k, S_{k+1})$ denotes the immediate reward when taking action $A_k$ at state $S_k$ and arriving at state $S_{k+1}$, capital letters denote random variables and lowercase letters denote values. The conventional objective function $J(\theta)$ of reinforcement learning [36] takes the following form

$$J(\theta) \triangleq \mathbb{E}_{S_0, A_0}[Q^{\pi_\theta}(S_0, A_0)] = \mathbb{E}_{\tau \sim \pi} \left[ R(\tau) \cdot P(\tau | \pi_\theta) \right], \quad (2)$$

where $\tau$ is a trajectory, i.e., $\tau = (S_0, A_0, \cdots)$, and

$P(\tau | \pi_\theta) = d_0(s_0) \cdot \prod_{k=0}^{T} \mathbb{P}(s_{k+1} | s_k, a_k) \pi_\theta(a_k | s_k).$

The Bellman equation [36] converts (1) into a recursive form as follows

$$Q^\pi(s, a) = \sum_{s' \in \mathcal{S}} \mathbb{P}(s' \mid s, a) \left[ R(s, a, s') + \gamma \sum_{a' \in \mathcal{A}} \pi(s', a') Q^\pi(s', a') \right]$$
$$= R(s, a) + \gamma \sum_{s' \in \mathcal{S}} \mathbb{P}(s' \mid s, a) \sum_{a' \in \mathcal{A}} \pi(s', a') Q^\pi(s', a'), \quad (3)$$

which expresses the expected reward as a summation of immediate reward $R(s, a)$ and discounted future rewards, and the immediate reward $R(s, a)$ is defined as $R(s, a) = \sum_{s' \in \mathcal{S}} \mathbb{P}(s' \mid s, a) R(s, a, s')$.

The Bellman's optimality equation [36] is

$$Q^*(s, a) = \sum_{s' \in \mathcal{S}} \mathbb{P}(s' \mid s, a) \left[ R(s, a, s') + \gamma \max_{a'} Q^*(s', a') \right]. \quad (4)$$

The optimal policy $\pi^*$ is given by a greedy strategy such that $\pi^* = \arg\max_\pi Q^\pi(s, a)$.

## 2.2 Observational Experiments for Multiple Policies and Physically Stationary Policy

As we pointed out in Fig. 1, theoretically there exists multiple policies. Here, we perform observational experiments on four challenging MuJoCo tasks [39], namely, Humanoid, Hopper, HalfCheetah, and Ant (details given in Appx. B.1), which are typical examples of the locomotion control of a robot.

We render the obtained policies over multiple runs and then identify physically stationary ones. We observe various types of moving strategies, as shown in Fig. 2, which verifies that multiple policies

Table 1: Analogy between MDP and quantum K-spin Ising model.

| MDP (Our formulation in (7)) | | Quantum K-spin Ising Model [26, 12] in (5) | |
| --- | --- | --- | --- |
| State-action pairs | $\mu_0, ..., \mu_{K-1}$ | Spins | $j_0, \cdots, j_{K-1}$ |
| Optimal policy | $\pi^*_{\mu_0} \times \pi^*_{\mu_1} \times \cdots \times \pi^*_{\mu_{K-1}}$ | Optimal configuration | $\sigma_{j_0} \times \sigma_{j_1} \times \cdots \times \sigma_{j_{K-1}}$ |
| Policy | $\pi_{\mu_0} \times \pi_{\mu_1} \times \cdots \times \pi_{\mu_{K-1}}$ | Configuration | $\sigma_{j_0} \times \sigma_{j_1} \times \cdots \times \sigma_{j_{K-1}}$ |
| Discounted reward | $L_{\mu_0...\mu_{K-1}}$ | Density function | $L_{j_0...j_{K-1}}$ |
| Functional of policy | $H(\pi_{\mu_0}, ..., \pi_{\mu_{K-1}})$ | Functional of spins | $H(\sigma_{j_0}, \cdots, \sigma_{j_{K-1}})$ |
| Stationary condition | $\frac{\delta H(\pi_{\mu_0}, \cdots, \pi_{\mu_{K-1}})}{\delta \pi_\mu} = 0$ | Stationary condition | $\frac{\delta H(\sigma_{j_0}, \cdots, \sigma_{j_{K-1}})}{\delta \sigma_j} = 0$ |

are very common. For example, the Humanoid agent learns either jumping with a single leg or running with two legs, as shown in Fig. 2 (top-left); another interesting example is HalfCheetah, in which an agent can run normally or in a flipped manner, as shown in Fig. 2 (bottom-left). Among the obtained policies, one can easily identify the physically stationary polices that control the robot moving forward with a *stable gait* (defined as gait that does not lead to fall).

## 3 Reinforcement Learning as Quantum K-spin Hamiltonian Equation

First, we make a novel analogy between the MDP and a quantum K-spin Ising model, and reformulate the objective function (2) into a quantum K-spin Hamiltonian equation. Then, we show that this Hamiltonian equation helps fix the issue of multiple fixed points in Fig. 1.

### 3.1 Motivation through Analogy with Quantum K-spin Ising Model

Given the multiple policies in Section 2.2, it is critical to employ a proper criteria to search for a physically stationary policy. In Table 1, we make a novel analogy between an MDP and a quantum K-spin Ising model [26, 12].

The Hamiltonian equation for a quantum K-spin Ising model [26, 12] measures the energy of a particular configuration of a quantum K-spin system and takes the following form

$$H = -\sum_{k=0}^{K-1} \sum_{j_0=1}^{N} \cdots \sum_{j_k=1}^{N} L_{j_0...j_k} \sigma_{j_0} \cdots \sigma_{j_k}, \tag{5}$$

where $N$ is the number of spins in the $k$-th configuration, $\sigma_{j_k} = \pm 1$ are spin variables, and $L_{j_0...j_k}$ is an energy density function for $k$ nearest spins' configuration $(\sigma_{j_0}, \ldots, \sigma_{j_k})$.

**Analogy in Table 1**. Starting from an analogy between a state-action pair $\mu_k = (S_k, A_k)$ and a spin $j_k$, we can map an optimal policy $\pi^*(\mu_k) \in \{0, 1\}$ in (4) to a single-qubit spin operator $\sigma_{j_k} \in \{-1, 1\}$ via $\pi^*(\mu_k) \longleftrightarrow (\mathbb{1}_{\mu_k} - \sigma_{\mu_k})/2$, where $\pi_\theta(\mu_k)$ denotes the probability of taking action $A_k$ at state $S_k$, following policy $\pi_\theta$. The energy density function $L_{j_0...j_k}$ can be defined as the discounted reward on a path $(\mu_0, \cdots, \mu_{k-1})$ of length $k$, i.e.,

$$L_{\mu_0,...,\mu_k} = \gamma^k \cdot R(\mu_k) \cdot d_0(s_0) \cdot \prod_{\ell=0}^{k-1} \mathbb{P}(s_{\ell+1}|\mu_\ell), \quad \text{(obtained via Monte Carlo simulation)} \tag{6}$$

where $d_0(s_0)$ denotes the distribution of initial state $s_0$. Analogy to the quantum K-spin Ising model, we can derive a functional of policy $H(\pi_{\mu_0}, ..., \pi_{\mu_{K-1}})$ in the context of MDP, which measures the energy of an RL policy. In this case, a physically stationary policy satisfies the stationary condition $\delta H(\pi_{\mu_0}, \cdots, \pi_{\mu_{K-1}})/\delta \pi_\mu = 0$.

## 3.2 Reformulation into Quantum K-spin Hamiltonian Equation

Inspired by [17], we formally reformulate (2) into a $K$-spin Hamiltonian equation

$$
\begin{aligned}
H(\theta) &\triangleq -\mathbb{E}_{S_0, A_0}\left[Q^{\pi_\theta}(S_0, A_0)\right] \\
&= -\lim_{K \to \infty} \sum_{k=0}^{K-1} \sum_{\mu_0}^{\mathcal{S} \times \mathcal{A}} \cdots \sum_{\mu_k}^{\mathcal{S} \times \mathcal{A}} L_{\mu_0, \ldots, \mu_k} \pi_\theta(\mu_0) \cdots \pi_\theta(\mu_k), \\
&= -\lim_{K \to \infty} \mathbb{E}_{\mu_0, \mu_1, \ldots, \mu_K}\left[\sum_{k=0}^{K-1} L_{\mu_0, \ldots, \mu_k}\right],
\end{aligned}
\tag{7}
$$

~~where $K \to \infty$~~, the expectation is taken over $S_0 \sim d_0(\cdot)$, $A_0 \sim \pi_\theta(S_0, \cdot)$, and the density function $L_{\mu_0, \ldots, \mu_k}$ is given in (6).

The following blue part will be put into the appendix.

**Monte Carlo Estimator** [30]: Consider a general probabilistic objective $\mathcal{F}$ of the form:

$$
\mathcal{F} \triangleq \mathbb{E}_{p(\boldsymbol{x};\theta)}[f(\boldsymbol{x}; \phi)],
\tag{8}
$$

in which a function $f$ of an input variable $\boldsymbol{x}$ with *structural parameters* $\phi$ is evaluated on average with respect to an input distribution $p(\boldsymbol{x}; \theta)$ with *distributional parameters* $\theta$.

A Monte Carlo method evaluates the function by first drawing independent samples $\hat{\boldsymbol{x}}^{(1)}, \ldots, \hat{\boldsymbol{x}}^{(N)}$ from the distribution $p(\boldsymbol{x}; \theta)$, and then computing the average:

$$
\widehat{\mathcal{F}}_N = \frac{1}{N} \sum_{i=1}^{N} f(\hat{\boldsymbol{x}}^{(i)}), \text{ where } \hat{\boldsymbol{x}}^{(i)} \sim p(\boldsymbol{x}; \theta) \text{ for } i = 1, \ldots, N.
\tag{9}
$$

The Monte Carlo estimator for (2) is

$$
\widehat{J}(\theta) = \frac{1}{N} \sum_{i=1}^{N} R(\tau^{(i)}), \text{ where } \tau^{(i)} \sim P(\tau^{(i)}|\pi_\theta) \text{ for } i = 1, \ldots, N,
\tag{10}
$$

and

$$
P(\tau^{(i)}|\pi_\theta) = d_0(s_0^{(i)}) \cdot \prod_{k=0}^{T} \mathbb{P}(s_{k+1}^{(i)}|s_k^{(i)}, a_k^{(i)}) \pi_\theta(a_k^{(i)}|s_k^{(i)}).
\tag{11}
$$

The Monte Carlo estimator for (7) is

$$
\widehat{H}(\theta) = \frac{1}{N'} \sum_{i=1}^{N'} \sum_{k=0}^{K-1} L_{\mu_0^{(i)}, \ldots, \mu_k^{(i)}}, \text{ for } i = 1, \ldots, N',
\tag{12}
$$

and

$$
L_{\mu_0^{(i)}, \ldots, \mu_k^{(i)}} = \gamma^k \cdot R(\mu_k^{(i)}) \cdot d_0(s_0^{(i)}) \cdot \prod_{\ell=0}^{k-1} \mathbb{P}(s_{\ell+1}^{(i)}|\mu_\ell^{(i)}).
\tag{13}
$$

**Remark**: The above two Monte Carlo estimators are quite different in the simulation process. (11) samples a random trajectory by following an environment's stochastic transition and a policy. In contrast, (13) measures a random path's discounted reward (the "energy") without following any policy, and the Hamiltonian equation (7) combinatorially enumerates all possible paths of length $K$ over the state-action space. In other words, the simulation process of the Hamiltonian term does not rely on any policy. Therefore, the Hamiltonian term is a suitable regularizer for both on-policy and off-policy algorithms.

This fundamental difference is due to the Ising model in (5), which combinatorially enumerates all paths and separates the environment and the policy.

148 **Physical interpretation**: Analogy to a quantum K-spin system, $H(\theta)$ in (7) measures the energy of
149 policy $\pi$. We hypothesize that the energy of a policy is a favorable criteria, since a stationary policy
150 with minimum energy: 1). *achieves a relative high reward independent of the initialization*; and 2). *is*
151 *robust to interference/noise in the inference stage*.

152 **K-step truncation in practice**. Minimizing (7) is NP-hard [11]. Since $\gamma \in (0, 1)$, $\gamma^K$ monotonically
153 decreased with look-ahead steps $K$, therefore, we truncate (7) to finite $K$ terms. One can show
154 that these $K$ terms in (7) is a geometric sequence with a truncation error ratio $1 - \gamma^K$. Assuming
155 $1 - \gamma^K \leq 1 - \epsilon$, where $\epsilon > 0$ is small, thus we have the look-ahead steps satisfies $K \geq \log_\gamma \epsilon$.

### 3.3 Revisiting Examples in Fig. 1

157 We elaborate how adding the energy measured by (7) onto each state can help drive to the terminal
158 state (a stationary policy), which fixes the foundational issue of multiple fixed points in Fig. 1 where
159 $\gamma = 1$. We have $H(0) = 0$ for the terminal state 0.

160 • (a) **Shortest path problem (deterministic)**: $H(1) = -\sum_{k=1}^{\infty} b = -\infty$. At state 1, the Bellman's
161   optimality equation becomes $V(1) = \max\{V(1) + \lambda H(1), b\}$. Independent of the initial value
162   $V_0(1)$, an agent obtains a policy that always transits back to terminal state 0.

163 • (b) **Blackmailer's problem (stochastic)**: $H(1) = -\infty$. The Bellman's optimality equation
164   becomes $V(1) = \max_a\{a + (1 - a)(V(1) + \lambda H(1))\}$ for state 1. For any $V_0(1) < \infty$, the optimal
165   policy becomes $a = 1$ that drives to the terminal state 0.

166 • (c) **Optimal stopping problem (terminating policies)**: any policy that takes infinite steps will
167   have $H(x) = -\infty$, since at each step number $k$, there are always trajectories that jump to point 0
168   with reward $-c$; and a direct jumping policy will have $H(x) = -c$. Therefore, adding $H(x)$ to
169   each point $x \neq 0$ will lead to a policy of *jumping back to point* 0.

## 4 Actor-Critic Algorithm with Quantum K-spin Hamiltonian Regularization

171 We propose a generic actor-critic algorithm with a H-term, derive two Hamiltonian's policy gradient
172 theorems for both deterministic and stochastic cases, and present the Monte Carlo gradient estimation.

### 4.1 Stationary Actor-Critic Algorithm with H-term

174 Actor-critic algorithms in reinforcement learning perform a bilevel optimization, namely alternating
175 between approximating a value function and optimizing a policy. In practice, a critic network with
176 parameter $\phi$ approximates the $Q$-value function, and an actor network with parameter $\theta$ approximates
177 the policy $\pi$, details given in Appx. F. However, since the critic's update is governed by the Bellman's
178 optimality equation, actor-critic algorithms suffer the multiple fixed points problem.

179 Motivated by Section 3.3, we propose a novel H-term for both deterministic and stochastic actor-critic
180 algorithms. Similar to the entropy term in [23], the proposed H-term is an add-on term to regularize
181 the actor network and help it converge to a stationary policy. Specifically, the objective functions of
182 actor and critic networks become:

$$
\begin{cases}
\text{Actor}: \max_\theta J_\pi(\theta, \phi) \triangleq (1 - \gamma)\mathbb{E}_{S_0 \sim d_0, A_0 \sim \pi_\theta(S_0, \cdot)}\left[Q_\phi\left(S_0, A_0\right)\right] - \lambda H(\theta), \\
\text{Critic}: \min_\phi J_Q(\theta, \phi) \triangleq \frac{1}{2}\mathbb{E}_{S \sim d_\theta(\cdot), A \sim \pi_\theta(S, \cdot)}\left[\left(Q_\phi(S, A) - y(S, A)\right)^2\right],
\end{cases} \tag{14}
$$

183 where a target Q-value is $y(S_k, A_k) = R(S_k, A_k) + \gamma Q_\phi(S_{k+1}, A_{k+1})$, and $\lambda > 0$ is a temperature
184 parameter. As an interpretation, the second term $-\lambda H(\theta)$ in the maximization objective function of
185 actor network aims to find a minimum energy configuration for the MDP problem, namely, a policy
186 $\pi$ that will add a minimum amount of energy to each state's value function (as in Section 3.3).

187 **New algorithm**. In Alg. 1, an agent interacts with an environment and alternatively updates its actor
188 network and critic network. The algorithm has $M$ episodes and each episode consists of a (Monte
189 Carlo) simulation process and a learning process (gradient estimation) as follows:

190 • During the (Monte Carlo) simulation process (lines 5-10 of Alg. 1), an agent takes action $a_t$ ac-
191   cording to a policy $\pi_\theta(\cdot|s_t)$, $t = 0, \cdots, T - 1$, generating a trajectory of $T$ steps/transitions.

---

**Algorithm 1** Stationary Actor-Critic Algorithm with H-term
---
1: **Input**: learning rate $\alpha$, temperature $\lambda$, look-ahead step $K$, and parameters $M, T, G, B, B'$
2: Initialize actor network $\pi$ and critic network $Q$ with parameters $\theta, \phi$, and replay buffers $\mathcal{D}_1, \mathcal{D}_2$
3: **for** episode $= 1, \cdots, M$ **do**
4:     Initialize state $s_0$
5:     **for** $t = 0, \cdots, T - 1$ **do**
6:         Select action $a_t \sim \pi_\theta(\cdot | s_t)$
7:         Execute action $a_t$, receive reward $r_t$, and observe new state $s_{t+1}$
8:         Store a transition $(s_t, a_t, r_t, s_{t+1})$ in $\mathcal{D}_1$
9:     **end**
10:    Store a trajectory $\tau$ of length $T$ in $\mathcal{D}_2$
11:    **for** $g = 1, \cdots, G$ **do**
12:       Randomly sample a mini-batch of $B$ transitions $\{(s_i, a_i, r_i, s_{i+1})\}_{i=1}^B$ from $\mathcal{D}_1$
13:       Randomly sample a mini-batch of $B'$ trajectories (of length $K$) $\{\tau_j\}_{j=1}^{B'}$ from $\mathcal{D}_2$
14:       Update critic network using a conventional method
15:       Update actor network as $\theta \leftarrow \theta + \alpha \left( \nabla_\theta \widehat{J}(\theta) - \lambda \, \nabla_\theta \widehat{H}(\theta) \right)$.
16:    **end**
17: **end**
---

Then, these $T$ transitions are stored into a replay buffer $\mathcal{D}_1$, while the full trajectory $\tau = (s_0, a_0, r_0, s_1, \cdots, s_{T-1}, a_{T-1}, r_{T-1}, s_T)$ is stored in replay buffer $\mathcal{D}_2$.

• During the learning process ($G \geq 1$ updates in one episode) (lines 11-16 of Alg. 1), a mini-batch of $B$ transitions $\{(s_i, a_i, r_i, s_{i+1})\}_{i=1}^B$ and a mini-batch of $B'$ trajectories (of length $K$) $\{\tau_j = (s_0^j, a_0^j, r_0^j, s_1^j, \cdots, s_{K-1}^j, a_{K-1}^j, r_{K-1}^j, s_K^j)\}_{j=1}^{B'}$ are sampled from $\mathcal{D}_1$ and $\mathcal{D}_2$, respectively. The critic network is updated by a conventional method, e.g., minimizing the mean squared error (MSE) between an estimated Q-value and a target value. The actor is updated by a Monte Carlo gradient estimator over $B$ transitions and $B'$ trajectories.

**Two new hyperparameters**. We introduce two hyperparameters: a temperature $\lambda > 0$ that is a relative weight of the H-term, and a look-ahead step $K \leq T$ that defines the horizon of the H-term.

**Implementation of replay buffer $\mathcal{D}_2$**. After a full trajectory $\tau$ of length $T$ is generated, it is partitioned into $T - K + 1$ trajectories of length $K$. We rank them according to the cumulative reward and store the top portion, say $80\%$, into a new replay buffer $\mathcal{D}_2$ (line 10 of Alg. 1). We randomly sample a mini-batch of $B'$ trajectories from $\mathcal{D}_2$ (line 13 of Alg. 1) to compute the H-term.

### 4.2 Hamiltonian Policy Gradient and Monte Carlo-based Gradient Estimator

We provide the policy gradient of the quantum K-spin Hamiltonian equation in (7), which are variants of the well-known policy gradient theorem [36]. We provide detailed derivations in Appx. D.

**Stochastic version**. The Hamiltonian stochastic gradient of (7) w.r.t. parameter $\theta$ is

$$\nabla_\theta H(\theta) = -\mathbb{E}_{\mu_0, \ldots, \mu_{K-1}} \left[ \sum_{k=0}^{K-1} \gamma^k \cdot R(\mu_k) \cdot \nabla_\theta \log \left( \pi_\theta(\mu_0) \cdot \pi_\theta(\mu_1) \cdots \pi_\theta(\mu_k) \right) \right]. \quad (15)$$

**Deterministic version**. Let $\eta_\theta(\cdot) : \mathcal{S} \to \mathcal{A}$ denote a deterministic policy, while we use $\widetilde{\pi}_{\theta,\delta}(\mu)$ to represent that a Gaussian noise (a.k.a, an exploration noise) with standard deviation $\delta > 0$ is added in the exploration process. The Hamiltonian deterministic gradient of (7) w.r.t. parameter $\theta$ is

$$\nabla_\theta H'(\theta) = -\mathbb{E}_{\mu_0, \ldots, \mu_{K-1}} \left[ \sum_{k=0}^{K-1} \gamma^k \cdot R(\mu_k) \cdot \nabla_\theta \log \left( \widetilde{\pi}_{\theta,\delta}(\mu_0) \cdot \widetilde{\pi}_{\theta,\delta}(\mu_1) \cdots \widetilde{\pi}_{\theta,\delta}(\mu_k) \right) \right]. \quad (16)$$

The quantum K-spin Hamiltonian equation in (7) is a reformulation of (2). We verify the gradient calculation by showing that: when $K \to \infty$, the Hamiltonian stochastic and deterministic policy gradient $\nabla_\theta H(\theta)$ and $\nabla_\theta H'(\theta)$ are equal to the stochastic policy gradient $\nabla_\theta J(\theta)$ in [37] and deterministic policy gradient $\nabla_\theta J'(\theta)$ in [35], respectively.

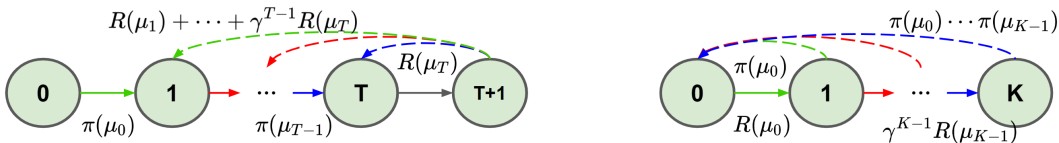

Figure 3: Comparison of REINFORCE's (left) and Hamiltonian's policy gradient (right).

Note that the gradient $\nabla_\theta H(\theta)$ in (15) and $\nabla_\theta H'(\theta)$ in (16) w.r.t. a distributional parameter $\theta$ takes an expectation form. Thus, a Monte Carlo gradient estimator is practically useful. We obtain the Monte Carlo gradient estimator of $\nabla_\theta H(\theta)$, illustrated in Fig. 3 (right), as follows

$$\nabla_\theta \widehat{H}(\theta) = -\frac{1}{N'} \sum_{i=1}^{N'} \left[ \sum_{k=0}^{K-1} \gamma^k \cdot R(\mu_k^i) \cdot \nabla_\theta \log \left[ \pi_\theta(\mu_0^i) \cdots \pi_\theta(\mu_k^i) \right] \right]. \tag{17}$$

As a contrast, we provide the Monte Carlo gradient estimator of REINFORCE's [37] policy gradient, as illustrated in Fig. 3 (left), as follows

$$\nabla_\theta \widehat{J}(\theta) = \frac{1}{NT} \sum_{i=1}^{N} \left[ \sum_{t=0}^{T-1} G_t^i \cdot \nabla_\theta \log \pi_\theta(\mu_t^i) \right], \quad \text{where } G_t^i = \sum_{t'=t+1}^{T} \gamma^{t'-t-1} R(\mu_{t'}^i). \tag{18}$$

An interesting observation is that both gradient calculations follow a similar pattern as shown in Fig. 3. REINFORCE's policy gradient [37] in Fig. 3 (left) employs an estimate of future rewards, while Hamiltonian policy gradient in Fig. 3 (right) uses trajectories in replay buffer $\mathcal{D}_2$.

**Computational complexity**: we measure the computation complexity by the times of computing one $\nabla_\theta \log \pi_\theta(\mu)$. Assume $N = B$ and $N' = B'$, since most DRL algorithms use a mini-batch stochastic gradient decent methods. REINFORCE's [37] policy gradient in (18) takes $O(BT)$ computations, while Alg. 1 adds $O(B'K(K+1)/2)$ computations in each gradient update step, thus a total complexity of $O(BT + B'K(K+1)/2)$.

## 5 Performance Evaluation

We evaluate the proposed H-term from four aspects: 1) increasing cumulative reward, 2) reducing variance, 3) driving to physically stationary policy, and 4) the impact of trajectory length $K$. All experiments were executed on an NVIDIA DGX-2 server [10]. The server contains 8 A100 GPUs, 320 GB GPU memory, and 128 CPU cores running at 2.25 GHz.

### 5.1 Experimental Settings

**Environments (tasks)**. We consider six challenging MuJoCo tasks [39] as in Section 2.2. The agent learns to control the locomotion of a robot and aims to move forward as quickly as possible. These tasks have high-dimensional continuous state and action spaces, in which there exists multiple locally optimal polices as revealed in Section 2.2.

**Compared algorithms**. To evaluate deterministic and stochastic algorithms, we choose Deep Deterministic Policy Gradient (DDPG) [27] and Proximal Policy Optimization (PPO) [34]. Since the H-term is compatible with existing variance reduction techniques, we implement the PPO algorithm with GAE [33]. For a fair comparison, we keep the hyperparameters (listed in Appx. H) the same and make sure that the obtained results reproduce existing benchmark tests [13].

**Performance metrics**. We employ two performance metrics, the cumulative rewards and variance, while in Section 5.4, we further consider different policies and report the number of convergence. We run each experiment with 20 random seeds and in each run we test 100 episodes.

### 5.2 H-term Increases Cumulative Reward

Experience replay is crucial in improving performance in terms of cumulative reward. The proposed H-term in (17) can be viewed as a novel experience replay technique for an actor network. Here, we

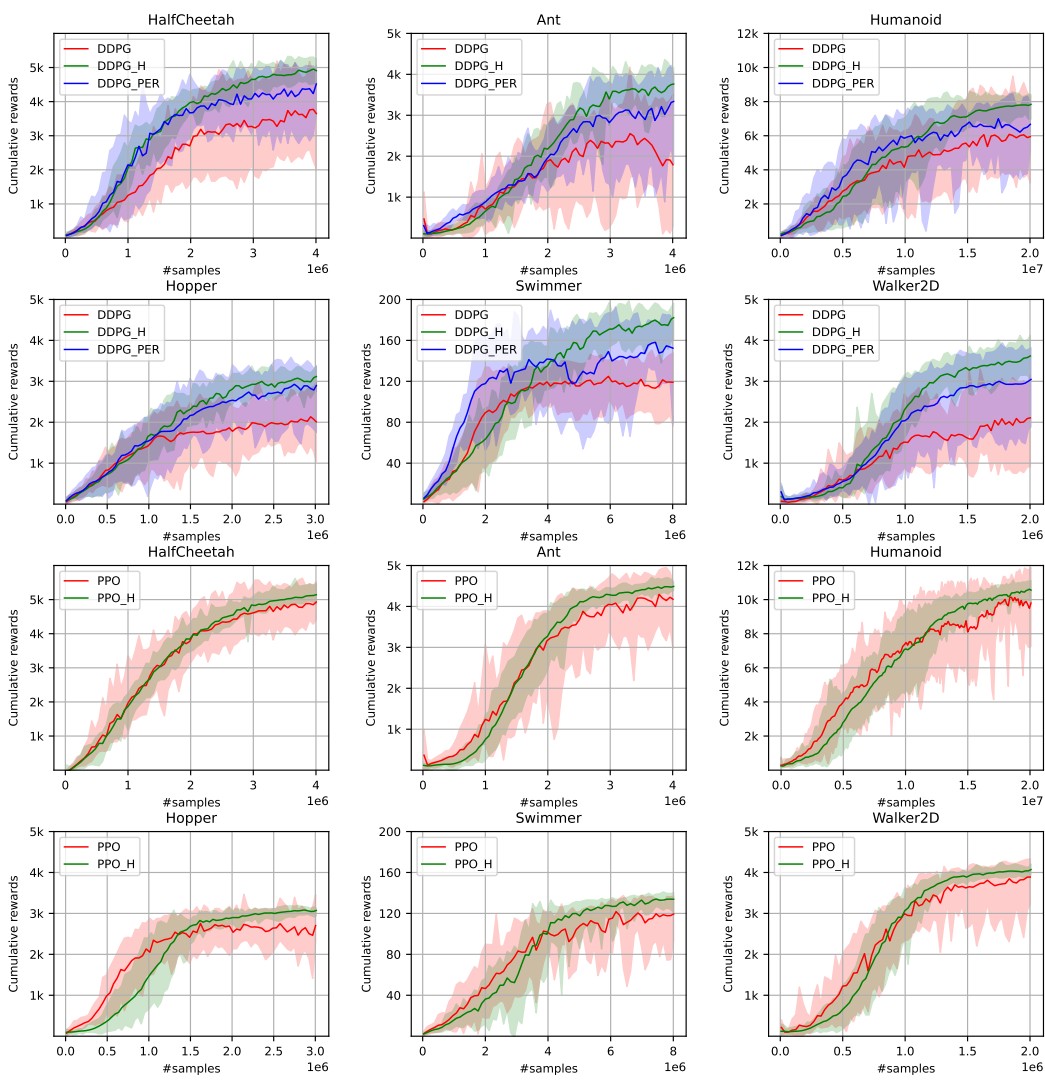

Figure 4: Cumulative rewards vs. #samples for compared DRL algorithms on six MuJoCo tasks.

add a compared algorithm, DDPG with Prioritized Experience Replay [32] (DDPG+PER), where PER prioritizes experience by the TD error to update a critic network.

In Fig. 4, both DDPG+PER and DDPG+H achieve a substantial improvement of cumulative reward. In particular, DDPG+H achieves the highest cumulative rewards in all six tasks, which are comparable to PPO's performance in Fig. 4. It is worthwhile to discuss the advantage of DDPG+H over DDPG+PER. DDPG+PER utilizes a prioritized replay strategy to obtain a more accurate critic network, however, it is updated via the Bellman equation with the trouble of multiple fixed points. In contrast, the H-term in DDPG+H is performed on the actor network. Our results indicate that an experience replay technique on actor network may be much more powerful.

## 5.3 H-term Reduces Variance

The PPO algorithm with GAE is regarded as the state-of-the-art algorithm in MuJoCo environments. However, it still has a very high variance (the shaded area) after the policies have converged, as shown in Fig. 4. We observe that, at the end of training, the PPO algorithm has a variance of 969.2, 1563.4, 2513.5, 905.3, 60.7, 1290.1 in the six tasks, respectively. Such a high variance is mainly due to the fact that the agent may converge to a random one of multiple policies.

Table 2: Experimental results on six challenging MuJoCo tasks.

| Tasks | Policies | PPO | | PPO+H ($K=8$) | | PPO+H ($K=16$) | | PPO+H ($K=24$) | |
|---|---|---|---|---|---|---|---|---|---|
| HalfCheetah | **running** | **13** | 4720.8 ±969.2 | **19** | 5028.4 ±211.3 | **20** | 5104.3 ±228.4 | **20** | 4995.1 ±383.3 |
| | flipping | 5 | | 0 | | 0 | | 0 | |
| | diving | 1 | | 1 | | 0 | | 0 | |
| | balancing | 1 | | 0 | | 0 | | 0 | |
| Ant | **running** | **17** | 4164 ±1563.4 | **20** | 4505.3 ±253.6 | **20** | 4645.6 ±225.4 | **20** | 4662.5 ±277.5 |
| | jumping | 0 | | 0 | | 0 | | 0 | |
| | flipping | 3 | | 0 | | 0 | | 0 | |
| Humanoid | **two-legs** | **7** | 9433.4 ±2513.5 | **17** | 9670.3 ±497.2 | **16** | 10189.1 ±683.7 | **17** | 9942.2 ±538.4 |
| | one-leg | 12 | | 3 | | 4 | | 3 | |
| | backward | 1 | | 0 | | 0 | | 0 | |
| Hopper | **hopping** | **10** | 2659.3 ±905.3 | **18** | 3116.5 ±289.4 | **20** | 3300.1 ±184.2 | **20** | 3340.7 ±191.5 |
| | diving | 8 | | 2 | | 0 | | 0 | |
| | balancing | 2 | | 0 | | 0 | | 0 | |
| Swimmer | **moving** | **14** | 110.7 ±60.7 | **20** | 130.6 ±33.5 | **20** | 132.5 ±31.6 | **19** | 132.2 ±36.2 |
| | balancing | 6 | | 0 | | 0 | | 1 | |
| Walker | **walking** | **5** | 5461.7 ±1290.1 | **16** | 5819.9 ±315.6 | **16** | 5927.2 ±296.8 | **15** | 6089.3 ±314.7 |
| | diving | 8 | | 2 | | 4 | | 5 | |
| | balancing | 7 | | 2 | | 0 | | 0 | |

In Fig. 4, the shaded areas of PPO+H ($K=16$) are dramatically smaller, i.e., a variance of 228.4, 225.4, 683.7, 184.2, 31.6, and 296.8, respectively. The variance has been reduced by $65.2\% \sim 85.6\%$, which verifies the effectiveness of the proposed H-term. In Fig. 4, we also observe that the H-term can help the DDPG algorithm reduce variances, namely, the variances of DDPG+H are much smaller than those of vanilla DDPG and DDPG+PER. Therefore, we may conclude that the H-term guides the agent to search for a stationary policy among multiple feasible ones.

More experimental results are given in Appx. H due to the space limit, including the cases of $K=8$ and $K=24$, and the curves of the H-term value during the training process. One may verify that the stationary policies have relative lower H-values.

## 5.4 H-term Drives to Physically Stationary Policy

A key question needs to be answered: *is H-term really guiding the agent converge to a physically stationary policy?* Similar to Section 2.2, we perform observational experiments on the six MuJoCo tasks and measure the number of convergence to the different policies over 20 runs. As shown in Table 2, the vanilla PPO algorithm converges to the physically stationary policy (**bold**) with 13, 17, 7, 10, 14, and 5 times for the six tasks, while the PPO+H ($K=16$) converges to the stationary policy with 20, 20, 16, 20, 20, and 16 times, respectively. From the empirical observation, we find that the PPO gets stuck in locally optimal policies, failing to find a consistent one. As expected, PPO+H can converge to the stationary policy with a substantially higher ratio, which verifies the effectiveness of the proposed H-term in finding a physically stationary policy.

## 5.5 Impact of Trajectory Length $K$

We investigate the impact of trajectory length $K$ that defines the horizon of the H-term. Based on (17), we foresee that a large $K$ means a more accurate estimation of $\nabla_\theta H(\theta)$ but at a price of computations. Here, we evaluate the PPO+H with $K=8, 16$ and set the size of replay buffer $\mathcal{D}_2$ to $1,000$. In Table 2, we observe that the cumulative reward increases and the variance decreases as $K$ increases from 8 to 16. However, for the case $K=24$, both metrics get worse due to the out-of-memory issue. For $K=24$, we reduce the replay buffer size to 800 to meet the memory limit. The smaller replay buffer size hurts the diversity of the trajectories and may lead to a small performance drop. This hypothesize is verified in Appx. H.2, where all trials use a consistent replay buffer size 800.

## 6   Conclusions

In this paper, we have addressed a foundational issue of existing deep reinforcement learning algorithms that **Bellman's optimality equation has multiple fixed points**. This issue leads to the instability of DRL algorithms, puts a challenge on their reliability and reproducibility, and thus limits the wider adoption in real-world tasks. As a fix of the problem, we propose a novel H-term, a physically inspired regularizer, by making a novel analogy between a MDP and a quantum K-spin Ising model. Experimentally, we show that the H-term helps DRL algorithms find a stationary policy that has the lowest energy in a system and reduces the variance of cumulative rewards by $65.2\% \sim 85.6\%$ compared with those of existing algorithms.

For future works, we will explore the potential of directly training a policy network using (7) as in Appx. I, quantum simulator [25] and quantum reinforcement learning [5][16]. It is interesting to apply Monte Carlo estimator for unbiased policy gradient calculations. We would like to show that the proposed H-term can help distributional RL algorithms [3] find a stationary policy, since the distributional Bellman optimality operator is not a contraction and thus there is also no unique policy.

## Broader Impact Statement

In the field of DRL research, this paper points out the issue of multiple fixed points and may attract more attention from various perspectives. In terms of applications, this paper moves toward the reliable and reproducible research by improving the stability of existing DRL algorithms. On the other hand, the obtained stationary policy may have broad practical impact in many real-world application areas, including but not limited to robotics, transportation, and finance.

From an interdisciplinary perspective, our approach lies at the intersection of (quantum) Hamiltonian mechanics and DRL, espcially the explicit analogy between an MDP and a quantum K-spin Ising model. We hope that our study will attract more attention to bring together the strengths of both approaches and yield new insights in both fields.

In terms of broader societal impact of this work, we do not see any foreseeable strongly negative impacts. However, this work essentially makes a tradeoff between the computational resource and stability, which may lead to future works with higher computational cost.

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

| (a) Shortest path problem | (b) Blackmailer's problem | (c) Optimal stopping problem |

Figure 5: Revisiting Fig. 1 for the discounted cases where $\gamma \in (0, 1)$.

## A    More Examples with Multiple Fixed Points

First, we consider the discounted formulations of the three examples (shown in Fig. 1), as shown in Fig. 5 where $\gamma \in (0, 1)$. The differences are marked in red.

- (a) **Shortest path problem (deterministic, discounted case)**: Given two states 1 and 0, an agent at state 1 transits to either state 1 or 0 with rewards $r = c$ or $r = b$, respectively. $c > (1 - \gamma) \cdot b$. At state 0, the value function is $V(0) = 0$. At state 1, the Bellman's optimality equation is $V(1) = \max\{c + \gamma \cdot V(1), b\}$, where any $V(1) \geq (b - c)/\gamma$ is a solution. If initialize $V_0(1) \geq b$, an agent obtains a policy that always transits back to state 1; otherwise, a result policy drives to terminal state 0.

- (b) **Blackmailer's problem (stochastic, discounted case)**: Different from (a), a profit maximizing blackmailer/agent at 1 demands a cash amount $a \in (0, 1]$ (an action), while a victim transits to state 1 with probability $a$ or to state 0 with probability $1 - a$, respectively. At state 0, a victim always refuses to yield to the blackmailer's demand, i.e., $V(0) = 0$. The Bellman's optimality equation is $V(1) = \max_a\{a + \gamma \cdot (1 - a)V(1)\}$ for state 1, where any $V(1) \geq 1$ is a feasible solution. If initialize $V_0(1) = c > 1$, the blackmailer's policy is demanding $a \to 0$ at the $k$-th step to keep the victim stay at state 1, for any $k \leq K_0 = -\lfloor \log_\gamma c \rfloor$; and taking $a = 1$ to transit to terminal state 0 at the $k$-th step, for any $k \geq K_0 + 1$; otherwise initialize $V_0(1) = c \leq 1$, the result policy is demanding the maximum $a = 1$ that drives the victim to a refusal state 0 (a terminal state).

- (c) **Optimal stopping problem (terminating policies, discounted case)**: In a space $\mathbb{R}^2$ with terminating state at point 0, at point $x \neq 0$ an agent moves to either point 0 with negative reward $-c$ or point $\alpha x$ with reward $-\|x\|$, respectively, where $\alpha \in (0, 1)$. The Bellman's optimality equation is $V(x) = \max\{-c, -\|x\| + \gamma \cdot V(\alpha x)\}$ and the optimal policy is to continue inside the sphere of radius $(1 - \alpha)c$ and to stop outside. If add a cone region $C$ within which an agent always receives a reward $-c$, a second policy is jumping to point 0 at any point in region $C$.

Then, we elaborate how the proposed H-term fixes the problems in Fig. 5.

### (a) Shortest path problem (deterministic, discounted case)

Assume $V_0(1) \geq b$ and $c > (1 - \gamma)b$, we have

$$V_1(1) = c + \gamma \cdot V_0(1) \geq c + \gamma \cdot b > b$$
$$V_2(1) = c + \gamma \cdot c + \gamma^2 \cdot V_0(1) \geq (1 + \gamma)c + \gamma^2 b > b$$
$$V_3(1) = c + \gamma \cdot c + \gamma^2 c + \gamma^3 \cdot V_0(1) \geq (1 + \gamma + \gamma^2)c + \gamma^3 b > b$$
$$\cdots$$

$$V_k(1) = \sum_{i=0}^{k-1} \gamma^i \cdot c + \gamma^k \cdot V_0(1) \geq \sum_{i=0}^{k-1} \gamma^i \cdot c + \gamma^k b > b \tag{19}$$

$$\cdots$$

$$V^*(1) = \sum_{i=0}^{\infty} \gamma^i \cdot c = \frac{1}{1 - \gamma} c > b$$

The values of $H(0)$ and $H(1)$ are as follows:

$$H(0) = 0, \quad H(1) = -b - \sum_{k=2}^{\infty} (\sum_{i=1}^{k-1} \gamma^{i-1} \cdot c + \gamma^k b) = -\infty. \tag{20}$$

Adding the above H-values to state 1 and 0, respectively, we have

$$V^*(1) + H(1) = \sum_{i=0}^{\infty} \gamma^i \cdot c - \infty = -\infty$$
$$V^*(0) + H(0) = b. \tag{21}$$

Therefore, $V^*(1) + H(1) < V^*(0) + H(0)$, independent of the initial value $V_0(1)$. That is, an agent always obtains a policy that drives to terminal state 0 at step 1.

(b) **Blackmailer's problem (stochastic, discounted case)**

If initialize $V_0(1) = c > 1$, the blackmailer's policy is demanding $a \to 0$ at the $k$-th step to keep the victim stay at state 1, for any $k \le K_0 = -\lfloor \log_\gamma c \rfloor$; and taking $a = 1$ to transit to terminal state 0 at the $k$-th step, for any $k \ge K_0 + 1$; otherwise initialize $V_0(1) = c \le 1$, the result policy is demanding the maximum $a = 1$ that drives the victim to a refusal state 0 (a terminal state).

The values of $H(0)$ and $H(1)$ are as follows:

$$H(0) = 0, \quad H(1) = -\sum_{k=1}^{\infty} \sum_{i=1}^{k-1} \gamma^{i-1} \cdot a = -\infty. \tag{22}$$

For arbitrary initial value of $V_0(1)$, $V_1(1) = a + (1-a) \cdot \gamma (V_0(1) + H(1))$ take maximum $V_1(1) = 1$ when $a = 1$. Therefore, the policy always drives to terminal state 0 at step 1.

(c) **Optimal stopping problem (terminating policies, discounted case)**

Any policy that takes infinite steps will have

$$H(x) = -c - \sum_{k=2}^{\infty} \left[ \sum_{i=1}^{k-1} \gamma^i \cdot \alpha^i \cdot \|x\| + \gamma^k \cdot (-c) \right] = -\infty \tag{23}$$

and a direct jumping policy will have $H(x) = -c$. Therefore, the H-term drives to a terminating policy.

## B MuJoCo Tasks with Multiple Policies

### B.1 Description of MuJoCo Taks

We selected six challenging robotic locomotion tasks from MuJoCo, namely, Swimmer-v3, Hopper-v3, Walker2D-v3, HalfCheetah-v3, Ant-v3, Humanoid-v3. Table 3 lists the action space and state space of each task.

Table 3: The state and action spaces of six challenging MuJoCo tasks.

| Tasks | Agent | Action Space | State Space |
|---|---|---|---|
| Swimmer-v3 | Three-link swimming robot | 2 | 8 |
| Hopper-v3 | Two-dimensional one-legged robot | 3 | 11 |
| Walker2d-v3 | Two-dimensional bipedal robot | 6 | 17 |
| HalfCheetah-v3 | Two-dimensional robot | 6 | 17 |
| Ant-v3 | Four-legged creature | 8 | 111 |
| Humanoid-v3 | Three-dimensional bipedal robot | 17 | 376 |

### B.2 Multiple policies in MuJoCo tasks

In the supplementary files, we includes rendered videos of different policies, as given in Table 4.

- Different policies are obtained over 20 runs of the PPO algorithm. We rendered theses polices and classified them by physical gaits.
- The policies in bold texts are physically stationary.

Table 4: List of video files for different policies.

| Task | Different Policies | Video Name |
|---|---|---|
| Hopper | **hopping** | hopper_hopping.mp4 |
| | diving | hopper_diving.mp4 |
| | standing | hopper_standing.mp4 |
| Ant | **running** | ant_running.mp4 |
| | standing | ant_standing.mp4 |
| | flipping | ant_flipping.mp4 |
| Walker | **walking** | walker_walking.mp4 |
| | diving | walker_diving.mp4 |
| | standing | walker_standing.mp4 |
| Humanoid | **two-legs** | humanoid_two_legs.mp4 |
| | one-leg | humanoid_one_leg.mp4 |
| | backward | humanoid_backward.mp4 |
| HalfCheetah | **running** | halfcheetah_running.mp4 |
| | diving | halfcheetah_diving.mp4 |
| | flipping | halfcheetah_flipping.mp4 |
| | standing | halfcheetah_standing.mp4 |
| Swimmer | **moving** | swimmer_moving.mp4 |
| | standing | swimmer_standing.mp4 |

# C   Quantum K-Spin Hamiltonian Formulation of Reinforcement Learning

516  We provide the detailed steps of reformulating (1) into a $K$-spin Hamiltonian equation

$$
\begin{aligned}
H(\theta) &\triangleq -\mathbb{E}_{S_0,A_0}\left[Q^{\pi_\theta}(S_0,A_0)\right] \\
&= -\mathbb{E}_{S_0,A_k\sim\pi_\theta(S_k,\cdot),S_{k+1}\sim\mathbb{P}(\cdot|S_k,A_k)}\left[\sum_{k=0}^{\infty}\gamma^k\cdot R(S_k,A_k)\right] \\
&= -\sum_{k=0}^{K-1}\mathbb{E}_{S_0,A_0,\cdots,S_k\sim\mathbb{P}(\cdot|S_{k-1},A_{k-1}),A_k\sim\pi_\theta(S_k,\cdot)}\left[\gamma^k\cdot R(S_k,A_k)\right] \\
&= -\sum_{k=0}^{K-1}\sum_{\mu_0}^{\mathcal{S}\times\mathcal{A}}\cdots\sum_{\mu_k}^{\mathcal{S}\times\mathcal{A}}\gamma^k\cdot R(\mu_k)\cdot d_0(S_0)\cdot\pi_\theta(\mu_0)\prod_{i=0}^{k-1}\left[\mathbb{P}(S_{i+1}|\mu_i)\cdot\pi_\theta(\mu_{i+1})\right] \\
&= -\sum_{k=0}^{K-1}\sum_{\mu_0}^{\mathcal{S}\times\mathcal{A}}\cdots\sum_{\mu_k}^{\mathcal{S}\times\mathcal{A}}\left[\gamma^k\cdot R(\mu_k)\cdot d_0(S_0)\cdot\prod_{i=0}^{k-1}\mathbb{P}(S_{i+1}|\mu_i)\right]\cdot\pi_\theta(\mu_0)\cdots\pi_\theta(\mu_k) \\
&= -\sum_{k=0}^{K-1}\sum_{\mu_0}^{\mathcal{S}\times\mathcal{A}}\cdots\sum_{\mu_k}^{\mathcal{S}\times\mathcal{A}}L_{\mu_0,\ldots,\mu_k}\cdot\pi_\theta(\mu_0)\cdots\pi_\theta(\mu_k),
\end{aligned}
\tag{24}
$$

517  where $K\to\infty$, and the density function is

$$
L_{\mu_0,\ldots,\mu_k} = \gamma^k\cdot R(\mu_k)\cdot d_0(S_0)\cdot\prod_{i=0}^{k-1}\mathbb{P}(S_{i+1}|\mu_i).
\tag{25}
$$

Table 5: Revisiting the analogy between MDP and quantum K-spin Ising model.

| MDP (Our formulation in (7)) | | Quantum K-spin Ising Model [26, 12] in (5) | |
|---|---|---|---|
| State-action pairs | $\mu_0,\ldots,\mu_{K-1}$ | Spins | $j_0,\cdots,j_{K-1}$ |
| Optimal policy | $\pi^*_{\mu_0}\times\pi^*_{\mu_1}\times\cdots\times\pi^*_{\mu_{K-1}}$ | Quantum field | $\sigma_{j_0}\times\sigma_{j_1}\times\cdots\times\sigma_{j_{K-1}}$ |
| Cumulative rewards | $L_{\mu_0\ldots\mu_{K-1}}$ | Density function | $L_{j_0\ldots j_{K-1}}$ |
| Functional of policy | $H(\pi_{\mu_0},\ldots,\pi_{\mu_{K-1}})$ | Functional of spins | $H(\sigma_{j_0},\cdots,\sigma_{j_{K-1}})$ |
| Stationary condition | $\frac{\delta H(\pi_{\mu_0},\cdots,\pi_{\mu_{K-1}})}{\delta\pi_\mu}=0$ | Stationary condition | $\frac{\delta H(\sigma_{j_0},\cdots,\sigma_{j_{K-1}})}{\delta\sigma_j}=0$ |

## D  Derivation Steps for Section 4.2: Hamiltonian's Policy Gradients

We provide the policy gradient of the quantum K-spin Hamiltonian equation in (7) for both stochastic and deterministic cases, which are variants of the well-known policy gradient theorem [36].

**Theorem 1.** *(**Hamiltonian's stochastic policy gradient**) The stochastic gradient of the K-spin Hamiltonian equation (7) w.r.t. parameter $\theta$ is*

$$\nabla_\theta H(\theta) = -\mathbb{E}_{\mu_0,\ldots,\mu_{K-1}} \left[ \sum_{k=0}^{K-1} \gamma^k \cdot R(\mu_k) \cdot \nabla_\theta \log \left( \pi_\theta(\mu_0) \cdot \pi_\theta(\mu_1) \cdots \pi_\theta(\mu_k) \right) \right]. \quad (26)$$

**Corollary 1.** *When $K \to \infty$, the Hamiltonian's stochastic policy gradient $\nabla_\theta H(\theta)$ in (26) is equal to the stochastic policy gradient $\nabla_\theta J(\theta)$ in [37],*

$$\lim_{K\to\infty} \nabla_\theta H(\theta) = -\nabla_\theta J(\theta) = -\mathbb{E}_{s\sim d_\theta, a\sim\pi_\theta} \left[ Q^{\pi_\theta}(s,a) \nabla_\theta \log \pi_\theta(s,a) \right]. \quad (27)$$

Let $\eta_\theta(\cdot) : \mathcal{S} \to \mathcal{A}$ denote a deterministic policy, while we use $\widetilde{\pi}_{\theta,\delta}(\mu)$ to represent that a Gaussian noise (a.k.a, an exploration noise) with standard deviation $\delta > 0$ is added in the exploration process.

**Theorem 2.** *(**Hamiltonian's deterministic policy gradient**) The deterministic gradient of the K-spin Hamiltonian equation (7) w.r.t. parameter $\theta$ is*

$$\nabla_\theta H'(\theta) = -\mathbb{E}_{\mu_0,\ldots,\mu_{K-1}} \left[ \sum_{k=0}^{K-1} \gamma^k \cdot R(\mu_k) \cdot \nabla_\theta \log \left( \widetilde{\pi}_{\theta,\delta}(\mu_0) \cdot \widetilde{\pi}_{\theta,\delta}(\mu_1) \cdots \widetilde{\pi}_{\theta,\delta}(\mu_k) \right) \right]. \quad (28)$$

**Corollary 2.** *When $K \to \infty$, the Hamiltonian's deterministic policy gradient $\nabla_\theta H'(\theta)$ in (28) is equal to the deterministic policy gradient $\nabla_\theta J'(\theta)$ in [35],*

$$\lim_{K\to\infty} \nabla_\theta H'(\theta) = -\nabla_\theta J'(\theta) = -\mathbb{E}_{s\sim d_\theta} \left[ \nabla_a Q^{\widetilde{\pi}_{\theta,\delta}}(s,a)|_{a=\eta_\theta} \nabla_\theta \eta_\theta(s) \right]. \quad (29)$$

**Corollary 3.** *When the variance of the exploration noise approaches zero, i.e., $\delta \to 0$, the deterministic policy gradient $\nabla_\theta H'(\theta)$ is the limiting case of the stochastic policy gradient $\nabla_\theta H(\theta)$,*

$$\nabla_\theta H'(\theta) = \lim_{\delta\to 0} \nabla_\theta H(\theta). \quad (30)$$

### D.1  Proof of Theorem 1: Hamiltonian's Stochastic Policy Gradient

*Proof.*

$$\nabla_\theta H(\theta) = -\sum_{k=0}^{K-1} \sum_{\mu_0}^{\mathcal{S}\times\mathcal{A}} \cdots \sum_{\mu_k}^{\mathcal{S}\times\mathcal{A}} L_{\mu_0,\ldots,\mu_k} \nabla_\theta \left[ \pi_\theta(\mu_0) \cdots \pi_\theta(\mu_k) \right]$$

$$= -\sum_{k=0}^{K-1} \sum_{\mu_0}^{\mathcal{S}\times\mathcal{A}} \cdots \sum_{\mu_k}^{\mathcal{S}\times\mathcal{A}} L_{\mu_0,\ldots,\mu_k} \left[ \pi_\theta(\mu_0) \cdots \pi_\theta(\mu_k) \right] \nabla_\theta \log \left[ \pi_\theta(\mu_0) \cdots \pi_\theta(\mu_k) \right]$$

$$= -\sum_{k=0}^{K-1} \sum_{\mu_0}^{\mathcal{S}\times\mathcal{A}} \cdots \sum_{\mu_k}^{\mathcal{S}\times\mathcal{A}} \gamma^k \cdot R(\mu_k) \cdot d_0(S_0) \cdot \pi_\theta(\mu_0) \prod_{i=0}^{k-1} \left[ \mathbb{P}(S_{i+1}|\mu_i) \cdot \pi_\theta(\mu_{i+1}) \right] \cdot \nabla_\theta \log \left[ \pi_\theta(\mu_0) \cdots \pi_\theta(\mu_k) \right]$$

$$= -\mathbb{E}_{\mu_0,\ldots,\mu_{K-1}} \left[ \sum_{k=0}^{K-1} \gamma^k \cdot R(\mu_k) \cdot \nabla_\theta \log \left[ \pi_\theta(\mu_0) \cdots \pi_\theta(\mu_k) \right] \right],$$

$$\quad (31)$$

where $\mu_k = (S_k, A_k)$, $S_0 \sim d_0(\cdot)$, $A_k \sim \pi_\theta(S_k, \cdot)$, $S_{k+1} \sim \mathbb{P}(\cdot \mid S_k, A_k)$ for $k = 0 \cdots K$. $\quad \square$

## D.2  Proof of Corollary 1

*Proof.*

$$
\nabla_\theta H(\theta) \overset{(a)}{=} -\sum_{k=0}^{K-1} \sum_{\mu_0}^{\mathcal{S}\times\mathcal{A}} \cdots \sum_{\mu_k}^{\mathcal{S}\times\mathcal{A}} L_{\mu_0,\ldots,\mu_k} \nabla_\theta \left[\pi_\theta(\mu_0)\cdots\pi_\theta(\mu_k)\right]
$$

$$
\overset{(b)}{=} -\sum_{k=0}^{K-1} \sum_{\mu_0}^{\mathcal{S}\times\mathcal{A}} \cdots \sum_{\mu_k}^{\mathcal{S}\times\mathcal{A}} L_{\mu_0,\ldots,\mu_k} \sum_{i=0}^{k} \pi_\theta(\mu_0)\cdots\pi_\theta(\mu_{i-1})\pi_\theta(\mu_{i+1})\cdots\pi_\theta(\mu_k)\nabla_\theta\pi_\theta(\mu_i)
$$

$$
\overset{(c)}{=} -\sum_{k=0}^{K-1} \sum_{\mu_0}^{\mathcal{S}\times\mathcal{A}} \cdots \sum_{\mu_k}^{\mathcal{S}\times\mathcal{A}} \gamma^k \cdot R(\mu_k) \cdot d_0(S_0) \prod_{i=0}^{k-1} \mathbb{P}(S_{i+1}|\mu_i) \sum_{i=0}^{k} \left[ \prod_{j=0}^{i-1} \pi_\theta(\mu_j) \cdot \nabla_\theta\pi_\theta(\mu_i) \cdot \prod_{j=i+1}^{k} \pi_\theta(\mu_j) \right]
$$

$$
\overset{(d)}{=} -\sum_{k=0}^{K-1} \sum_{\mu_0}^{\mathcal{S}\times\mathcal{A}} \cdots \sum_{\mu_k}^{\mathcal{S}\times\mathcal{A}} \sum_{i=0}^{k} d_0(S_0) \left[ \gamma^i \prod_{j=0}^{i-1} \pi_\theta(\mu_j)\mathbb{P}(S_{j+1}|\mu_j) \right] \nabla_\theta\pi_\theta(\mu_i) \left[ \prod_{j=i+1}^{k-1} \pi_\theta(\mu_j)\mathbb{P}(S_{j+1}|\mu_j)\pi_\theta(\mu_k)\gamma^{k-i}R(\mu_k) \right]
$$

$$
\overset{(e)}{=} -\sum_{k=0}^{K-1} \sum_{i=0}^{k} \sum_{S_0}^{\mathcal{S}} d_0(S_0) \sum_{S_i}^{\mathcal{S}} \rho(S_0,S_i,i) \sum_{A_i}^{\mathcal{A}} \nabla_\theta\pi_\theta(S_i,A_i) \cdot \sum_{\mu_k}^{\mathcal{S}\times\mathcal{A}} \rho(S_i,S_k,k-i) \cdot \pi_\theta(\mu_k) \cdot R(\mu_k)
$$

$$
\overset{(f)}{=} -\sum_{S_0}^{\mathcal{S}} d_0(S_0) \sum_{S}^{\mathcal{S}} \sum_{i=0}^{K-1} \rho(S_0,S,i) \sum_{A}^{\mathcal{A}} \nabla_\theta\pi_\theta(S,A) \cdot \left[ \sum_{S'}^{\mathcal{S}} \sum_{k=i}^{K-1} \rho(S,S',k-i) \cdot \sum_{A'}^{\mathcal{A}} \pi_\theta(S',A') \cdot R(S',A') \right]
$$

$$
\overset{(g)}{=} -\sum_{S_0}^{\mathcal{S}} d_0(S_0) \sum_{S}^{\mathcal{S}} \sum_{i=0}^{\infty} \rho(S_0,S,i) \sum_{A}^{\mathcal{A}} \nabla_\theta\pi_\theta(S,A) \cdot Q^{\pi_\theta}(S,A)
$$

$$
\overset{(h)}{=} -\left[ \sum_{S}^{\mathcal{S}} \sum_{S_0}^{\mathcal{S}} d_0(S_0) \sum_{i=0}^{\infty} \rho(S_0,S,i) \right] \cdot \sum_{S}^{\mathcal{S}} \frac{\sum_{S_0}^{\mathcal{S}} d_0(S_0) \sum_{i=0}^{\infty} \rho(S_0,S,i)}{\sum_{s}^{\mathcal{S}} \sum_{S_0}^{\mathcal{S}} d_0(S_0) \sum_{i=0}^{\infty} \rho(S_0,S,i)} \sum_{A}^{\mathcal{A}} \nabla_\theta\pi_\theta(S,A) \cdot Q_\theta(S,A)
$$

$$
\overset{(i)}{\propto} -\sum_{S}^{\mathcal{S}} d_{\pi_\theta}(S) \sum_{A}^{\mathcal{A}} \nabla_\theta\pi_\theta(S,A) \cdot Q^{\pi_\theta}(S,A)
$$

$$
\overset{(j)}{=} -\mathbb{E}_{S\sim d_\theta, A\sim\pi_\theta(S,\cdot)}[Q^{\pi_\theta}(S,A)\nabla_\theta\log\pi_\theta(S,A)],
$$

$$\tag{32}$$

where $\rho(S,S',i)$ denotes the probability of state $S$ transfer to $S'$ in $i$ steps.

We provide detailed explanations step-by-step:

- Equality $(a)$ holds by definition.

- In equality $(b)$, using the chain rule, we take derivative of $\nabla_\theta[\pi_\theta(\mu_0)\cdots\pi_\theta(\mu_k)]$ with respect to $\pi_\theta(\mu_i)$, $i=1,\ldots,k$.

- In equality $(c)$, we plug in $L_{\mu_0,\cdots,\mu_k}$ in (6).

- In equality $(d)$, we insert $\mathbb{P}(S_{i+1}|\mu_i)\,\mathbb{P}(S_{i+1}|\mu_i)$ between $\pi_\theta(\mu_i)$ and $\pi_\theta(\mu_{i+1})$, $i=1,\ldots,k$.

- In equality $(e)$, we split trajectory $\mu_0,\cdots,\mu_i,\cdots,\mu_k$ into two trajectories $\mu_0,\cdots,\mu_i$ and $\mu_i,\cdots,\mu_k$. Therefore, we can classify all trajectories $\mu_0,\cdots,\mu_k$ by $\mu_0,\mu_i,\mu_k$, and $i$.

- In equality $(f)$, we reorganize $\sum_{k=0}^{K-1}\sum_{i=0}^{k}$ into $\sum_{i=0}^{K-1}\sum_{k=i}^{K-1}$. The former one first traverses the length $k$ of a trajeoctory, and then traverses the $i$-th step on it.The latter one first traverses the $i$-th step of a trajectory, and then traverses the length $k$ of it.

- In equality $(g)$, we calculate the limit of $(f)$ when $K$ approaches $\infty$.

- In equality $(h)$, we normalize $\sum_{S_0}^{\mathcal{S}} d_0(S_0) \sum_{i=0}^{\infty} \rho(S_0,S,i)$ to be a probability distribution.

- In equality $(i)$, we remove the constant $\sum_S^{\mathcal{S}} \sum_{S_0}^{\mathcal{S}} d_0(S_0) \sum_{i=0}^{\infty} \rho(S_0, S, i)$ and replace the fraction with $d_{\pi_\theta}(S)$, the stationary distribution of state $S$ under policy $\pi_\theta$.

- In equality $(j)$, we reformulate $(i)$ as expectation.

$\square$

## D.3  Proof of Theorem 2: Hamiltonian's Deterministic Policy Gradient

*Proof.* Let $\eta_\theta(\cdot) : \mathcal{S} \to \mathcal{A}$ denote a deterministic policy, while we use $\widetilde{\pi}_{\theta,\delta}(\mu)$ to represent that a Gaussian noise (a.k.a, an exploration noise) with standard deviation $\delta > 0$ is added in the exploration process. In the inference stage, there is no exploration noise, the policy is deterministic, i.e., $\delta = 0$ and $A_k = \eta_\theta(S_k)$.

$$
\begin{aligned}
H'(\theta) &\triangleq -\mathbb{E}_{S_0 \sim d_0, A_0 \sim \widetilde{\pi}_{\theta,\delta}} \left[ Q^{\widetilde{\pi}_{\theta,\delta}}(S_0, A_0) \right] \\
&= -\mathbb{E}_{S_0, A_k \sim \widetilde{\pi}_{\theta,\delta}(S_k, \cdot), S_{k+1} \sim \mathbb{P}(\cdot|S_k, A_k)} \left[ \sum_{k=0}^{\infty} \gamma^k \cdot R(S_k, A_k) \right] \\
&= -\sum_{k=0}^{K} \mathbb{E}_{S_0, A_k \sim \widetilde{\pi}_{\theta,\delta}(S_k, \cdot), S_{k+1} \sim \mathbb{P}(\cdot|S_k, A_k)} \left[ \gamma^k \cdot R(S_k, A_k) \right] \\
&= -\sum_{k=0}^{K} \sum_{\mu_0}^{\mathcal{S} \times \mathcal{A}} \cdots \sum_{\mu_k}^{\mathcal{S} \times \mathcal{A}} \gamma^k \cdot R(\mu_k) \cdot d_0(S_0) \cdot \widetilde{\pi}_{\theta,\delta}(\mu_0) \prod_{i=0}^{k-1} \left[ \mathbb{P}(S_{i+1}|\mu_i) \cdot \widetilde{\pi}_{\theta,\delta}(\mu_{i+1}) \right] \\
&= -\sum_{k=0}^{K} \sum_{\mu_0}^{\mathcal{S} \times \mathcal{A}} \cdots \sum_{\mu_k}^{\mathcal{S} \times \mathcal{A}} \left[ \gamma^k \cdot R(\mu_k) \cdot d_0(S_0) \cdot \prod_{i=0}^{k-1} \mathbb{P}(S_{i+1}|\mu_i) \right] \cdot \widetilde{\pi}_{\theta,\delta}(\mu_0) \cdots \widetilde{\pi}_{\theta,\delta}(\mu_k) \\
&= -\sum_{k=0}^{K} \sum_{\mu_0}^{\mathcal{S} \times \mathcal{A}} \cdots \sum_{\mu_k}^{\mathcal{S} \times \mathcal{A}} L_{\mu_0, \dots, \mu_k} \cdot \widetilde{\pi}_{\theta,\delta}(\mu_0) \cdots \widetilde{\pi}_{\theta,\delta}(\mu_k),
\end{aligned}
\tag{33}
$$

where $K \to \infty$, and

$$
L_{\mu_0, \dots, \mu_k} = \gamma^k \cdot R(\mu_k) \cdot d_0(S_0) \cdot \prod_{i=0}^{k-1} \mathbb{P}(S_{i+1}|\mu_i).
\tag{34}
$$

$\square$

### D.4 Proof of Corollary 2

*Proof.*

$$
\begin{aligned}
\nabla_\theta H'(\pi_\theta) &= -\sum_{k=0}^{K}\sum_{\mu_0}^{\mathcal{S}\times\mathcal{A}}\cdots\sum_{\mu_k}^{\mathcal{S}\times\mathcal{A}}\left(L_{\mu_0,\ldots,\mu_k}\cdot\nabla_\theta\left[\widetilde{\pi}_\theta(\mu_0)\cdots\widetilde{\pi}_\theta(\mu_k)\right]+\nabla_\theta L_{\mu_0,\cdots,\mu_k}\cdot\widetilde{\pi}_\theta(\mu_0)\cdots\widetilde{\pi}_\theta(\mu_k)\right)\\
&= -\sum_{k=0}^{K}\sum_{\mu_0}^{\mathcal{S}\times\mathcal{A}}\cdots\sum_{\mu_k}^{\mathcal{S}\times\mathcal{A}}\left[\widetilde{\pi}_\theta(\mu_0)\cdots\widetilde{\pi}_\theta(\mu_k)\right]\cdot\nabla_\theta L_{\mu_0,\ldots,\mu_k}\\
&= -\sum_{k=0}^{K}\sum_{\mu_0}^{\mathcal{S}\times\mathcal{A}}\cdots\sum_{\mu_k}^{\mathcal{S}\times\mathcal{A}}\nabla_\theta\left[\gamma^k\cdot R(\mu_k)\cdot d_0(S_0)\cdot\prod_{i=0}^{k-1}\mathbb{P}(S_{i+1}|\mu_i))\right]\\
&= -\sum_{k=0}^{K}\sum_{\mu_0}^{\mathcal{S}\times\mathcal{A}}\cdots\sum_{\mu_k}^{\mathcal{S}\times\mathcal{A}}\nabla_A\left[\gamma^k\cdot R(\mu_k)\cdot d_0(S_0)\cdot\prod_{i=0}^{k-1}\mathbb{P}(S_{i+1}|\mu_i))\right]\nabla_\theta\eta_\theta(S)\\
&= -\sum_{S_0}^{\mathcal{S}}d_0(S_0)\nabla_A\mathbb{E}_{S_{t+1}\sim\mathbb{P}(\cdot|S_t,A_t)}\left[\sum_{t=0}^{\infty}\gamma^k R(S_t,A_t)\right]\cdot\nabla_\theta\eta_\theta(S)\\
&= -\sum_{S_0}^{\mathcal{S}}d_0(S_0)\nabla_A Q(S_0,A_0)\cdot\nabla_\theta\eta_\theta(S)\\
&= -\mathbb{E}_{S_0\sim d_0(\cdot)}\left[\nabla_A Q(S_0,A_0)\cdot\nabla_\theta\eta_\theta(S)\right]
\end{aligned}
\tag{35}
$$

where $\mu_k=(S_k,A_k)$, $S_0\sim d_0(\cdot)$, $A_k\sim\pi_\theta(S_k,\cdot)$, $S_{k+1}\sim\mathbb{P}(\cdot\mid S_k,A_k)$, for $k=0\cdots K$. $\qquad\square$

### D.5 Proof of Corollary 3

*Proof.* In Corollary 2 and Corollary 1, we have

$$
\begin{aligned}
\nabla_\theta H'(\theta) &= -\nabla_\theta J'(\theta),\\
\nabla_\theta H(\theta) &= -\nabla_\theta J(\theta),
\end{aligned}
\tag{36}
$$

when $K\to\infty$.

[35] proved that

$$
\nabla_\theta J'(\theta) = \lim_{\delta\to 0}\nabla_\theta J(\theta),
\tag{37}
$$

where $\delta$ is the standard deviation of the Gaussian noise of stochastic policy $\pi_\theta$.

Therefore,

$$
\nabla_\theta H'(\theta) = \lim_{\delta\to 0}\nabla_\theta H(\theta)
\tag{38}
$$

$\square$

 # E  Variance Reduction (Newly Added)

For the general function in (8), one simple but effective variance reduction technique is to subtract a baseline term as follows:

$$\mathbb{E}_{p(\boldsymbol{x};\theta)}\left[(f(\boldsymbol{x}) - \beta)\nabla_\theta \log p(\boldsymbol{x};\theta)\right], \tag{39}$$

where $\beta$ is the baseline term.

**Our reasoning logic**:

1). We first briefly describe a high-level idea [19] that adding a baseline term, like the proposed H-term, will help reduce the gradient variance.

2). We sketch the steps to show how the proposed H-term will mathematically reduce the gradient variance, following the framework in Section 5.2 of [20].

**High-level IDEA**. One generic approach to reduce the variance of Monte Carlo estimates is to use an additive control variate. Suppose we wish to estimate the integral of the function $f : \mathcal{X} \to \mathbb{R}$, and we know the value of the integral of another function on the same space $\phi : \mathcal{X} \to \mathbb{R}$. We have

$$\int_\mathcal{X} f(x) = \int_\mathcal{X} (f(x) - \phi(x)) + \int_\mathcal{X} \phi(x), \tag{40}$$

and the integral of $f(x) - \phi(x)$ can be estimated. If $\phi(x) = f(x)$, meaning that , then we have managed to reduce our variance to zero [19]. More generally,

$$\mathrm{Var}(f - \phi) = \mathrm{Var}(f) - 2\mathrm{Cov}(f, \phi) + \mathrm{Var}(\phi). \tag{41}$$

If $\phi$ and $f$ are strongly correlated, so that the covariance term on the right hand side is greater than the variance of $\phi$, i.e., $-2\mathrm{Cov}(f, \phi) + \mathrm{Var}(\phi) \le 0$. then a variance reduction has been made over the original estimation problem [19], i.e., $\mathrm{Var}(f - \phi) \le \mathrm{Var}(f)$.

**Our reasoning**. Then, we present our reasoning.

Note that the gradient of the new objective function of the actor network in (14) consists of two components, namely $\nabla_\theta J(\theta)$ and $\nabla_\theta H(\theta)$. Here, we consider

$$\nabla_\theta J(\theta) - \lambda\, \nabla_\theta H(\theta), \quad \text{where } \lambda > 0 \text{ is a temperature parameter,} \tag{42}$$

where $\nabla_\theta J(\theta)$ in (27) is the above function $f(\cdot)$ and $\lambda\, \nabla_\theta H(\theta)$ in (15) is the above function $\phi(\cdot)$.

The Hamiltonian stochastic gradient in (15) has the optimal value

$$\nabla_\theta H^*(\theta) = -\lim_{K \to \infty} \mathbb{E}_{\mu_0,\ldots,\mu_{K-1}}\left[\sum_{k=0}^{K-1} \gamma^k \cdot R(\mu_k) \cdot \nabla_\theta \log\left(\pi_\theta(\mu_0) \cdot \pi_\theta(\mu_1) \cdots \pi_\theta(\mu_k)\right)\right]. \tag{43}$$

According to Theorem 8 of [19] that is proved via (41), we have

$$\mathrm{Var}\left[\nabla_\theta J(\theta) - \lambda\, \nabla_\theta H^*(\theta)\right] = \mathrm{Var}[\nabla_\theta J(\theta)] - \frac{1}{\lambda}\mathbb{E}_{s \sim d_\theta, a \sim \pi_\theta}\left[\frac{\left(\mathbb{E}_{s \sim d_\theta, a \sim \pi_\theta}\left[(\nabla_\theta \log \pi_\theta(s,a))^2 \nabla_\theta J(\theta)\right]\right)^2}{\mathbb{E}_{s \sim d_\theta, a \sim \pi_\theta}\left[(\nabla_\theta \log \pi_\theta(s,a))^2\right]}\right]$$
$$\le \mathrm{Var}[\nabla_\theta J(\theta)], \tag{44}$$

where the second term is positive, and

$$\nabla_\theta H^*(\theta) = \frac{\mathbb{E}_{s \sim d_\theta, a \sim \pi_\theta}\left(\left[\nabla_\theta \log \pi_\theta(s,a))^2 \nabla_\theta J(\theta)\right]\right)}{\mathbb{E}_{s \sim d_\theta, a \sim \pi_\theta}\left[(\nabla_\theta \log \pi_\theta(s,a))^2\right]}. \tag{45}$$

In Alg. 1 and Alg. 2, we used a general H-term $\nabla_\theta H(\theta)$, not the optimal one in (43). Next, we provide a general characterization for this case..

According to Theorem 10 of [19], we have

$$\mathrm{Var}\left[\nabla_\theta J(\theta) - \lambda\, \nabla_\theta H(\theta)\right] - \mathrm{Var}\left[\nabla_\theta J(\theta) - \lambda\, \nabla_\theta H^*(\theta)\right]$$
$$= \lambda^2\, \mathbb{E}_{s \sim d_\theta, a \sim \pi_\theta}\left[(\nabla_\theta \log \pi_\theta(s,a))^2 (\nabla_\theta H(\theta) - \nabla_\theta H^*(\theta))^2\right] \tag{46}$$

Assume Lipschiz continuity of the graident $\nabla_\theta H(\theta)$ such that

$$||\nabla_\theta H(\theta) - \nabla_\theta H^*(\theta)||_2 \leq 2L(H(\theta) - H^*(\theta)) \leq 2L\epsilon, \tag{47}$$

given $K \geq \log_\gamma \epsilon$ with $L > 0, \epsilon > 0$, as pointed out in the end of Section 3.2.

Therefore, combining (46), (48) with (48), we obtain that

$$\begin{aligned}
\mathrm{Var}\left[\nabla_\theta J(\theta) - \lambda \nabla_\theta H(\theta)\right] &= \mathrm{Var}[\nabla_\theta J(\theta)] - \frac{1}{\lambda}\mathbb{E}_{s\sim d_\theta, a\sim\pi_\theta}\left[\frac{\left(\mathbb{E}_{s\sim d_\theta, a\sim\pi_\theta}\left[(\nabla_\theta \log \pi_\theta(s,a))^2 \nabla_\theta J(\theta)\right]\right)^2}{\mathbb{E}_{s\sim d_\theta, a\sim\pi_\theta}\left[(\nabla_\theta \log \pi_\theta(s,a))^2\right]}\right] \\
&\quad + \lambda^2 (2L\epsilon)^2 \mathbb{E}_{s\sim d_\theta, a\sim\pi_\theta}\left[(\nabla_\theta \log \pi_\theta(s,a))^2\right] \\
&\leq \mathrm{Var}[\nabla_\theta J(\theta)],
\end{aligned} \tag{48}$$

when both $|\nabla_\theta \log \pi_\theta(s,a)|$ and $|\nabla_\theta J(\theta)|$ are upper bounded, e.g., $|\nabla_\theta \log \pi_\theta(s,a)| < C_1$ and $|\nabla_\theta J(\theta)| < C_2$; and we set $\epsilon, \lambda$ properly such that

$$\begin{aligned}
&-\frac{1}{\lambda}C_2^2 + 4\lambda^2 L^2 \epsilon^2 C_1^2 < 0 \\
&\lambda^3 \epsilon^2 < \frac{C_2^2}{4L^2 C_1^2},
\end{aligned} \tag{49}$$

which can be easily satisfied by properly selecting $\lambda$ and $K \geq \log_\gamma \epsilon$.

**Conclusion**:

To sum up, we show that it is easy to achieve $\mathrm{Var}\left[\nabla_\theta J(\theta) - \lambda \nabla_\theta H(\theta)\right] \leq \mathrm{Var}[\nabla_\theta J(\theta)]$, which means adding the H-term can lead to smaller variance than that of the conventional gradient.

## F    Conventional Actor-Critic Algorithms for Deep Reinforcement Learning

The gradient of (2) is [36]

$$\nabla_\theta J(\theta) \triangleq \sum_S^{\mathcal{S}} d_{\mathcal{S},\theta}(S) \sum_A^{\mathcal{A}} Q_\theta(S, A) \, \nabla_\theta \pi_\theta(S, A). \tag{50}$$

Since $Q_\theta$ in (50) is unknown [41] ( the stationary distribution $d_\theta$ is unknown), one can plug in a critic network with parameter $\phi$ as an estimator of $Q_\theta$ and obtain

$$\nabla_\theta^\phi J(\theta, \phi) = \sum_S^{\mathcal{S}} d_{\mathcal{S},\theta}(S) \sum_A^{\mathcal{A}} Q_\phi(S, A) \, \nabla_\theta \pi_\theta(S, A), \tag{51}$$

where $d_{\mathcal{S},\theta} \in \mathbb{R}_+^{|\mathcal{S}||\mathcal{A}| \times 1}$ denotes the stationary distribution over the states instead of state-action pairs.

(51) is a bi-level optimization problem [7], and a natural solution is an iterative algorithm that alternates between estimating $Q_\phi$ with parameter $\phi$ and improving policy $\pi_\theta$ with parameter $\theta$. Therefore, a family of actor-critic algorithms are proposed with following objective functions:

$$\begin{cases} \text{Actor} : \max_\theta J_\pi(\theta, \phi) = (1 - \gamma)\mathbb{E}_{S_0 \sim d_0, A_0 \sim \pi_\theta(S_0, \cdot)} \left[ Q_\phi(S_0, A_0) \right] \\ \text{Critic} : \max_\phi J_Q(\theta, \phi) = \dfrac{1}{2}\mathbb{E}_{S \sim d_\theta(\cdot), A \sim \pi_\theta(S, \cdot)} \left[ (Q_\phi(S, A) - y(S, A))^2 \right]. \end{cases} \tag{52}$$

The gradient of (52) can be estimated as follows

$$\begin{aligned} \nabla_\theta \widehat{J}_\pi(\theta, \phi) &= \frac{1}{N} \sum_{i=1}^{N} Q_\phi(\mu) \cdot \nabla_\theta \log \pi_\theta(\mu) \\ \nabla_\phi \widehat{J}_Q(\theta, \phi) &= \frac{1}{N} \sum_{i=1}^{N} [Q_\phi(S, A) - y(S, A)] \cdot \nabla_\phi Q_\phi(S, A) \end{aligned} \tag{53}$$

The parameters $\phi$ and $\theta$ are updated as follows:

$$\begin{cases} \text{Actor} : \quad \theta \leftarrow \theta + \alpha \, \nabla_\theta^\phi \widehat{J}_\pi, \\ \text{Critic} : \quad \phi \leftarrow \phi - \alpha \, \nabla_\phi \widehat{J}_Q. \end{cases} \tag{54}$$

# G   Stationary Deterministic Policy Gradient Algorithm with H-term

For completeness, we present the details of the deterministic actor-critic algorithm with H-term.

---
**Algorithm 2** Stationary Actor-Critic Algorithm with H-term

---
1: **Input**: learning rate $\alpha$, temperature $\lambda$, look-ahead step $K$, and parameters $\delta, M, T, G, B, B'$
2: Initialize actor network $\eta$ and critic network $Q$ with parameters $\theta, \phi$, and replay buffers $\mathcal{D}_1, \mathcal{D}_2$
3: **for** episode $= 1, \cdots, M$ **do**
4:     Initialize state $s_0$
5:     **for** $t = 0, \cdots, T-1$ **do**
6:         Take action $a_t = \eta_\theta(s_t) + \epsilon$, where $\epsilon \sim \mathcal{N}(0, \delta^2)$
7:         Execute action $a_t$, receive reward $r_t$, and observe new state $s_{t+1}$
8:         Store a transition $(s_t, a_t, r_t, s_{t+1})$ in $\mathcal{D}_1$
9:     **end**
10:    Store a trajectory $\tau$ of length $T$ in $\mathcal{D}_2$
11:    **for** $g = 1, \cdots, G$ **do**
12:       Randomly sample a mini-batch of $B$ transitions $\{(s_i, a_i, r_i, s_{i+1})\}_{i=1}^{B}$ from $\mathcal{D}_1$
13:       Randomly sample a mini-batch of $B'$ trajectories (of length $K$) $\{\tau_j\}_{j=1}^{B'}$ from $\mathcal{D}_2$
14:       Update critic network using a conventional method
15:       Update actor network as $\theta \leftarrow \theta + \alpha \left( \nabla_\theta \widehat{J}'(\theta) - \lambda \nabla_\theta \widehat{H}'(\theta) \right)$.
16:    **end**
17: **end**

---

We apply the proposed Hamiltonian equation (7) to regularize the actor network. Specifically, $H'(\theta)$ in (7) is added to the actor's objective with weight $\lambda > 0$. The objective functions of actor and critic networks become:

$$\begin{cases} \text{Actor}: \max_\theta J'_\pi(\theta, \phi) = (1-\gamma)\mathbb{E}_{S_0 \sim d_0, A_0 = \eta_\theta(S_0)} \left[ Q_\phi(S_0, A_0) \right] - \lambda H'(\theta), \\ \text{Critic}: \min_\phi J_Q(\theta, \phi) = \frac{1}{2}\mathbb{E}_{S \sim d_\theta(\cdot), A = \eta_\theta(S)} \left[ (Q_\phi(S, A) - y(S, A))^2 \right]. \end{cases} \tag{55}$$

The gradient of (55) is

$$\nabla_\theta J'_\pi(\theta, \phi) = (1-\gamma) \sum_S^{\mathcal{S}} d_{\mathcal{S}, \theta}(S) \nabla_A Q_\phi(S, A) \cdot \nabla_\theta \eta_\theta(S) - \lambda \nabla_\theta H'(\theta), \tag{56}$$

$$\nabla_\phi J_Q(\theta, \phi) = \sum_S^{\mathcal{S}} d_{S, \theta}(S) \cdot [Q_\phi(S, A) - y(S, A)] \cdot \nabla_\phi Q_\phi(S, A)|_{A = \eta_\theta(S)}. \tag{57}$$

To estimate $\nabla_\theta H'(\theta)$, the Monte Carlo gradient estimator in (17) is used. Therefore, (56) and (57) can be estimated as follows:

$$\nabla_\theta \widehat{J}'_\pi(\theta, \phi) = \frac{1}{N} \sum_{i=1}^{N} \left[ \nabla_A Q_\phi(S, A)|_{A = \eta_\theta(S)} \nabla_\theta \eta_\theta(S) \right] - \frac{1}{N'} \sum_{i=1}^{N'} \left[ \lambda \sum_{k=0}^{K} \gamma^k R(\mu_k) \nabla_\theta \log \left[ \widetilde{\pi}_\theta(\mu_0) \cdots \widetilde{\pi}_\theta(\mu_k) \right] \right], \tag{58}$$

$$\nabla_\phi \widehat{J}_Q(\theta, \phi) = \frac{1}{N} \sum_{i=1}^{N} [Q_\phi(S, A) - y(S, A)] \cdot \nabla_\phi Q_\phi(S, A)|_{A = \eta_\theta(S)}. \tag{59}$$

# H  Experiments: Hyperparameters and More Results

## H.1  Hyperparameters in Experiments

Table 6: Hyperparameters used for the PPO and PPO + H in MuJoCo tasks

| Parameters | Values |
|---|---|
| Optimizer | Adam |
| Learning rate | $3 \cdot 10^{-4}$ |
| Discount ($\gamma$) | 0.99 |
| GAE parameter | 0.95 |
| Number of hidden layers for all networks | 3 |
| Number of hidden units per layer | 256 |
| Mini-batch size | 32 |
| Importance rate of H-term ($\lambda$) | $2^{-3}$ |
| Truncation step of H-term (K) | 16 |

Table 7: Hyperparameters used for the DDPG and DDPG + H in MuJoCo tasks

| Parameters | Values |
|---|---|
| Optimizer | Adam |
| Learning rate | $5 \cdot 10^{-4}$ |
| Target Update Rate ($\tau$) | $10^{-3}$ |
| Discount ($\gamma$) | 0.995 |
| Replay buffer size | $10^{6}$ |
| Number of hidden layers for all networks | 3 |
| Number of hidden units per layer | 256 |
| Batch size | 64 |
| Importance rate of H-term ($\lambda$) | $2^{-3}$ |
| Truncation step of H-term (K) | 16 |

## H.2  More Results

Fig. 6 shows the H-value (average over 20 runs) during the training process, which verified that the trained agents have converged to policies with small H-values.

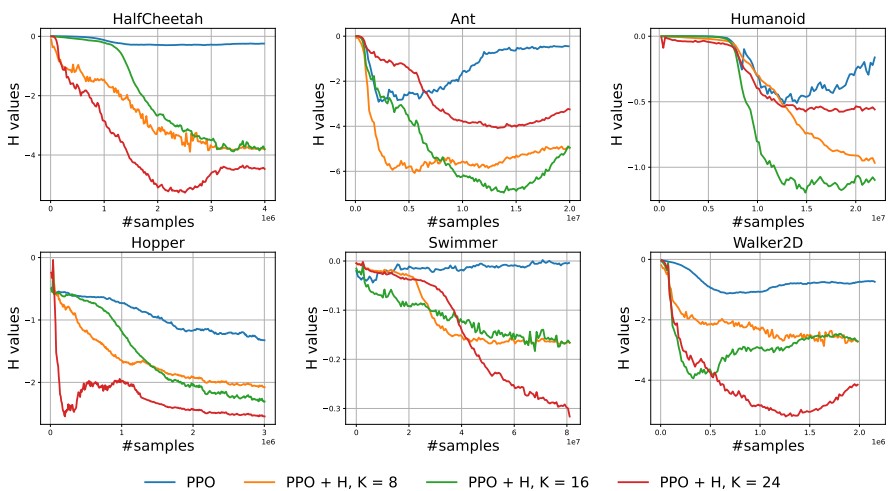

Figure 6: $H$ values during the training process.

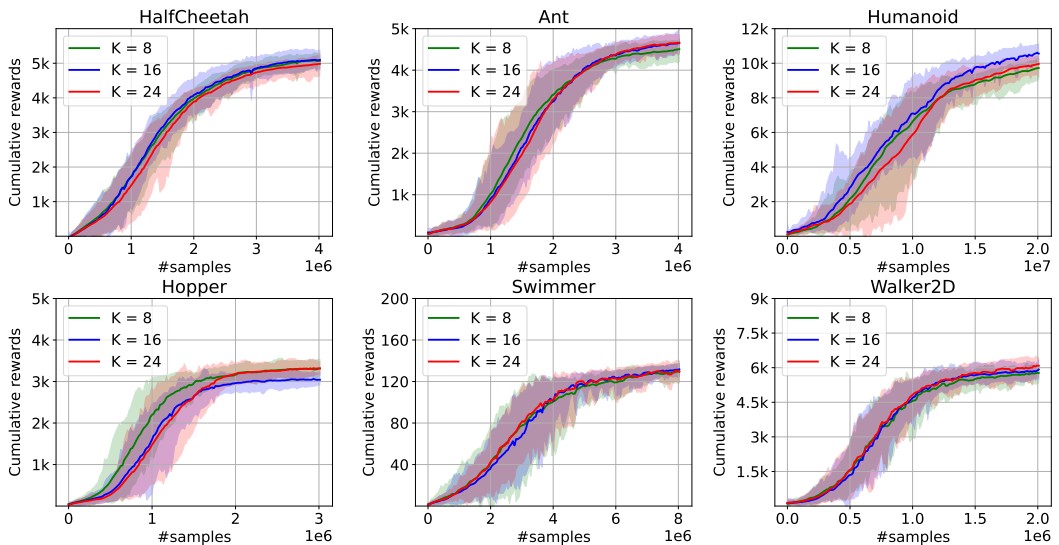

Figure 7: For the proposed PPO+H algorithm, the performance with different $K$ values.

| Tasks | Policies | | PPO | | PPO+H ($K = 8$) | | PPO+H ($K = 16$) | | PPO+H ($K = 24$) |
|---|---|---|---|---|---|---|---|---|---|
| HalfCheetah | **running** | **13** | 4720.8 ±969.2 | **17** | 4839.5 ±392.5 | **20** | 5001.8 ±321.5 | **20** | 4995.1 ±383.3 |
| | flipping | 5 | | 0 | | 0 | | 0 | |
| | diving | 1 | | 3 | | 0 | | 0 | |
| | balancing | 1 | | 0 | | 0 | | 0 | |
| Ant | **running** | **17** | 4164 ±1563.4 | **20** | 4351.6 ±294.5 | **20** | 4553.1 ±276.4 | **20** | 4662.5 ±277.5 |
| | jumping | 0 | | 0 | | 0 | | 0 | |
| | flipping | 3 | | 0 | | 0 | | 0 | |
| Humanoid | **two-legs** | **7** | 9433.4 ±2513.5 | **15** | 9583.4 ±753.4 | **15** | 9873.2 ±653.7 | **17** | 9942.2 ±538.4 |
| | one-leg | 12 | | 5 | | 5 | | 3 | |
| | backward | 1 | | 0 | | 0 | | 0 | |
| Hopper | **hopping** | **10** | 2659.3 ±905.3 | **18** | 3014.9 ±304.7 | **20** | 3254.2 ±246.1 | **20** | 3340.7 ±191.5 |
| | diving | 8 | | 2 | | 0 | | 0 | |
| | balancing | 2 | | 0 | | 0 | | 0 | |
| Swimmer | **moving** | **14** | 110.7 ±60.7 | **19** | 121.5 ±42.1 | **20** | 142.3 ±36.9 | **19** | 132.2 ±36.2 |
| | balancing | 6 | | 1 | | 0 | | 1 | |
| Walker | **walking** | **5** | 5461.7 ±1290.1 | **16** | 5794.4 ±341.8 | **16** | 5921.8 ±304.5 | **15** | 6089.3 ±314.7 |
| | diving | 8 | | 2 | | 3 | | 5 | |
| | balancing | 7 | | 2 | | 1 | | 0 | |

Table 8: Experimental results on six challenging MuJoCo tasks.

Fig. 7 shows more performance of the PPO+H algorithm, for $K = 8, 16, 24$. We run each experiment with 20 random seeds and each run we test 100 episodes.

To verify the hypothesize that smaller replay buffer hurts the performance, we rerun the trials of $K = 8, 16$ with a replay buffer size 800.

# I  Hamiltonian Policy Network

## I.1  Hamiltonian Policy Network

Since Hamiltonian equation in (7) is a functional of policy $\pi_\theta$, a natural question would be: can we use the Hamiltonian equation replace existing Bellman's equation (3) or the policy gradient's objective function (2)?

As a verification, we test the capability of Hamiltonian equation in (7) as a loss function to train a policy network. The algorithm is first given as follows.

---
**Algorithm 3** Hamiltonian Policy Network

---
1: **Input**: learning rate $\alpha$, look-ahead step $K$, and parameters $M, T, G, B$
2: Initialize policy network with parameters $\theta$, and replay buffer $\mathcal{D}$
3: **for** episode $= 1, \cdots, M$ **do**
4:     Initialize state $s_0$
5:     **for** $t = 0, \cdots, T-1$ **do**
6:         Select action $a_t \sim \pi_\theta(\cdot|s_t)$
7:         Execute action $a_t$, receive reward $r_t$, and observe new state $s_{t+1}$
8:     **end**
9:     Store a trajectory $\tau$ of length $T$ in $\mathcal{D}$
10:     **for** $g = 1, \cdots, G$ **do**
11:         Randomly sample a mini-batch of $B$ trajectories (of length $K$) $\{\tau_j\}_{j=1}^B$ from $\mathcal{D}$
12:         Update pocliy network as $\theta \leftarrow \theta - \alpha \, \nabla_\theta \widehat{H}(\theta)$.
13:     **end**
14: **end**

---

In Alg. 3, an agent interacts with an environment and updates its policy network. The algorithm has $M$ episodes and each episode consists of a (Monte Carlo) simulation process and a learning process (gradient estimation) as follows:

- During the (Monte Carlo) simulation process (lines 5-9 of Alg. 3), an agent takes action $a_t$ according to a policy $\pi_\theta(\cdot|s_t)$, $t = 0, \cdots, T-1$, generating a trajectory of $T$ steps/transitions. Then, the full trajectory $\tau = (s_0, a_0, r_0, s_1, \cdots, s_{T-1}, a_{T-1}, r_{T-1}, s_T)$ is stored in replay buffer $\mathcal{D}$.

- During the learning process ($G \geq 1$ updates in one episode) (lines 10-12 of Alg. 1), a mini-batch of $B$ trajectories (of length $K$) $\{\tau_j = (s_0^j, a_0^j, r_0^j, s_1^j, \cdots, s_{K-1}^j, a_{K-1}^j, r_{K-1}^j, s_K^j)\}_{j=1}^B$ are sampled from $\mathcal{D}$, respectively. The policy network is updated by a Monte Carlo gradient estimator over $B$ trajectories.

**Implementation of replay buffer** $\mathcal{D}$. After a full trajectory $\tau$ of length $T$ is generated, it is partitioned into $T - K + 1$ trajectories of length $K$. We rank them according to the cumulative reward and store the top portion, say $80\%$, into a new replay buffer $\mathcal{D}$ (line 9 of Alg. 3). We randomly sample a mini-batch of $B$ trajectories from $\mathcal{D}$ (line 11 of Alg. 3) to compute the H-term.

## I.2  Frozenlake Task

**Environment**: Frozenlake $8 \times 8$, a game in OpenAI Gym.

**Rules**: As shown in Fig. 8 (left), the Frozenlake task has $8 \times 8$ states with $4$ optional actions to move around. The agent needs to go from the start point and find the way to the destination in limited steps. There are $8$ holes which can cause the agent to fail the game.

**Experiment settings**: We take Deep Q-learning (DQN) [29] as our baseline and use the implementation from the ElegantRL library. We use a $4$-layer fully connected neural network as the deep policy network both in DQN and DHN. We use the Adam optimizer with a learning rate $1 \times 10^{-3}$ and a batch size $100$.

**Evaluation**: We evaluate the performance of policy by computing the success rate, in which we use $50$ agents to walk $100$ steps and compute the rates of agents who successfully arrive the destination.

**Results for the Frozenlake task**: Fig. 9 (left) shows the success rate of agents with increasing the number of transitions learned by the network. compared with DQN, DHN has a more stable training process. It is easy for DQN to quickly obtain a good policy to win the game. But with increasing the number of transitions fed to the network, the performance of DQN shows a large and frequent shock while the performance of DHN shows the strong stability.

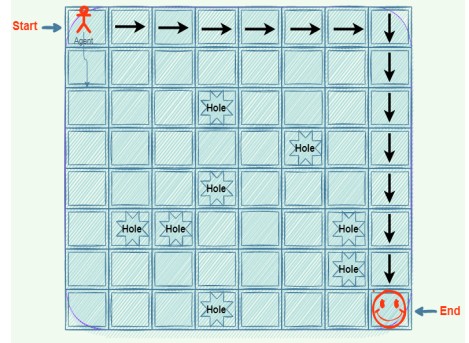 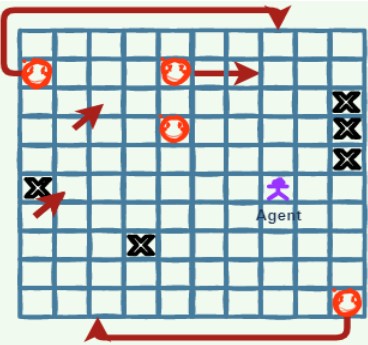

Figure 8: The Frozenlake task (left) and Gridworld task (right).

### I.3  Gridworld Task

**Environment**: a Gridworld of size $10 \times 10$, a game available in our code.

**Rules**: As shown in the Fig. 8 (right), the Gridworld has $10 \times 10$ states with $4$ optional actions to move around. The agent will initialize at a random locations and it needs to find the smiley as many as possible which has 10 reward in turn. It should be noted that there are some endpoints which may cause the agent game over and some transfer-points which transfer the agent to certain location.

**Experiment settings and evaluation**: Both the experiment settings and evaluation method are the same with that on Frozenlake $8 \times 8$ game.

**Results for the Gridworld task**: Fig. 9 (rigt) shows the mean reward obtained by the agents with increasing the training time. Compared with DQN, DHN has a faster training process. It only needs massive random parallel samples of trajectories and do not need any policy for guided sampling while DQN needs guided exploration in the training process which costs a large time consumption.

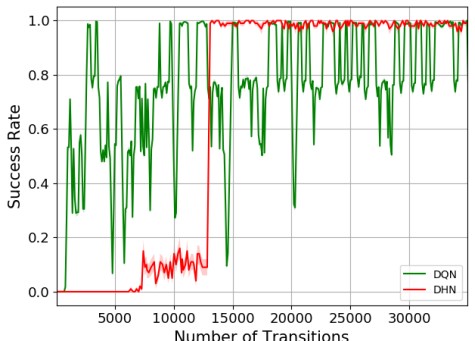 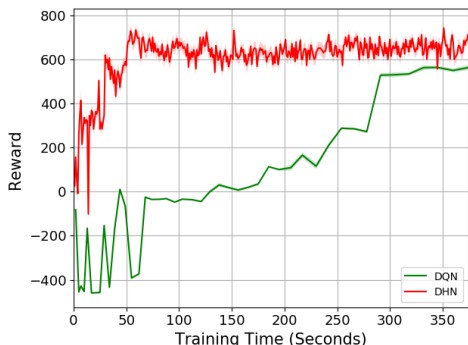

Figure 9: Comparison between the DQN and DHN algorithms. The Frozenlake task (left) and Gridworld task (right).

