# OpenReview forum: "Stationary Deep Reinforcement Learning with Quantum K-spin Hamiltonian Equation"
_NeurIPS.cc/2022/Conference — NeurIPS 2022 Submitted_

### Official Review · Reviewer_JucZ · 2022-07-03

**Rating:** 4
**Confidence:** 3
**Soundness:** 2 fair
**Presentation:** 3 good
**Contribution:** 2 fair

**Summary:**

This paper aims to resolve the challenge that Bellman's optimality equation has multiple fixed points, which leads to instability in deriving its solution. To this end, the authors first observe that the evaluation of cumulative reward function can be reformulated into a K-spin Ising model. The authors then propose to use the energy of such an Ising model as a regularizer in the actor-critic algorithm. The authors further conduct multiple experiments on the MuJoCo environment.

**Questions:**

$\textbf{Question 1.}$
Is the regularizer term in (7) equivalent to (2), namely, the corresponding cumulative reward function? Why does adding the cumulative reward function as a regularizer improves the stability of the actor-critic? In fact, by rewriting the log of products of policies into the sum of log policies and reorganizing the sum, I find the gradient equivalent with REINFORCE (with a truncation of trajectory to the $k$-th step). Can I understand the gradient of Hamiltonian as a variant of policy gradient estimation?

$\textbf{Question 2.}$
How does utilizing the Hamiltonian regularizer resolves the raised challenge that Bellman's optimality equation has multiple fixed points? Conjecturing that minimizing energy improves the stability is not sufficiently convincing as the formulation of Hamiltonian in equation (7) seems to be the same as the cumulative reward function. That said, minimizing the energy appears to be the same as maximizing the cumulative reward, which is also the objective of various existing policy gradient approaches. More importantly, the adopted AC-style approach is a policy gradient method, so it does not attempt to solve Bellman's optimality equation directly. In contrast, it improves the cumulative reward function by updating the policy directly.

**Limitations:**

N.A.

**Strengths And Weaknesses:**

$\textbf{Strength.}$

The quantum K-spin Ising model view of RL is interesting and novel to me.

$\textbf{Weakness.}$

The challenge raised by the authors does not seem to be fully addressed by the authors. See Question 2 for the details.

---

> ### Author Response · Authors · 2022-08-02
> **Response to Reviewer JucZ (1/2)**
>
> Thank you for your insightful feedback. We would like to address your concerns and answer your questions in the following.
>
> > Is the regularizer term in (7) equivalent to (2), namely, the corresponding cumulative reward function? Why does adding the cumulative reward function as a regularizer improves the stability of the actor-critic? In fact, by rewriting the log of products of policies into the sum of log policies and reorganizing the sum, I find the gradient equivalent with REINFORCE (with a truncation of trajectory to the $k$-th step). Can I understand the gradient of Hamiltonian as a variant of policy gradient estimation?
>
> The regularizer term in (7) is NOT exactly equivalent to (2) (details can be found in Appx. C, Equation (18)).
>
> * First, (2) is the cumulative reward function (the expectation is taken over trajectories); Equation (7) is derived from (1) which is the expectation of discounted rewards along a trajectory. We should have clearly specified $R(\tau)$ after (2).
> * Second, Fig. 3 made a comparison between REINFORCE’s policy gradient in (12) and the proposed Hamiltonian gradient in (11). Yes, the Hamiltonian gradient is a variant of policy gradient estimation.
> * Third, as shown in Fig. 3, we used a truncation of K steps in (7), and the discounted reward $L(\cdot)$ in (6) is calculated through Monte Carlo simulation.
> * An interesting point to make is that under the Ising model analogy of MDP/RL, the Hamiltonian equation has clear physical meaning, namely, it measures the “energy” of a policy. For this physically-inspired H-term, we derived a variant of the policy gradient estimation, as a regulation term (Alg. 1 in Line 15). Such an add-on term turns out to be rather simple to implement and delivers substantial performance improvements.

---

> > ### Author Response · Authors · 2022-08-02
> > **Response to Reviewer JucZ (2/2)**
> >
> > > How does utilizing the Hamiltonian regularizer resolves the raised challenge that Bellman's optimality equation has multiple fixed points? Conjecturing that minimizing energy improves the stability is not sufficiently convincing as the formulation of Hamiltonian in equation (7) seems to be the same as the cumulative reward function. That said, minimizing the energy appears to be the same as maximizing the cumulative reward, which is also the objective of various existing policy gradient approaches. More importantly, the adopted AC-style approach is a policy gradient method, so it does not attempt to solve Bellman's optimality equation directly. In contrast, it improves the cumulative reward function by updating the policy directly.
> >
> > The authors have to defend against this comment for several reasons.
> >
> > * First, from the above response, it is said that minimizing the energy by the Hamiltonian equation is NOT exactly maximizing the cumulative reward (the objective function of RL). Maybe the word “reformulate” caused some misunderstanding. However, they are quite similar, especially when the proposed H-term is an add-on term to regularize the policy network.
> > * Second, taking an optimization perspective, multiple fixed points mean multiple critical points (including saddle points and local minima); And Bellman's optimality equation having this issue means that each run with a different random initialization will randomly converge to one of many critical points, which the author believe is a fundamental cause of high variance (the current highly unstable DRL algorithms). Exploiting a term that measures the “energy” of the policy will provide a guide for the training process, which helps converge to critical points with lower energy.  Then, the problem becomes whether those critical points have a similar energy, or whether the Hamiltonian equation is a good metric. Since the Hamiltonian equation is universal for a lot of physical systems (robotic control, movements in gaming, etc.), the authors are confident.
> > * Some backup information on its ubiquity is:
> >     1. We found this phenomenon (randomly converging to one of many critical points, as shown in Fig. 2) in combinatorial search problems, e.g., graph max-cut, mixed integer learning programming, traveler salesman problem, and minimum independent cover;
> >     2. We even found it in resource allocation of 5G/6G wireless communication systems, e.g., power allocation beamformer design of MIMO base stations, respectively.
> > * Third, the adopted AC-style approach is a policy gradient method, which involves an estimate of Q-value (via Bellman's optimality equation). For RL, the Q-value estimation is a dual problem, while the primary problem is policy optimization (say via policy gradient). One claim of this work is that since Bellman’s optimality equation has the inherent issue of high variance (randomly converges to one of many critical points), we propose an add-on term to regularize the policy network. As shown by the experimental result in Section 5.2, we verify that prioritized experience replay (PER) on the policy network, achieved by the H-term, is better than PER on the critic network.
> > * In summary, the dual problem of Q-value estimation via Bellman's optimality equation is problematic itself, thus we are hoping this add-on H-term directly on the policy network can address a fundamental issue of DRL algorithms, namely how to reduce the variance of policies with different random seeds.

---

> > > ### Comment · Reviewer_JucZ · 2022-08-08
> > > **Response to the Review**
> > >
> > > Thanks for the response. I still find some points unclear to me.
> > >
> > > First, in (2) the cumulative reward is the expectation of $Q^{\pi_\theta}$ with respect to the initial state distribution, which I believe is exactly the same as (7) except for the negative sign.
> > >
> > >
> > > Second, PPO is an on-policy PG method that only solves the Bellman equation for the current policy. It does not solve the Bellman optimality equation.
> > >
> > >
> > > Finally, the argument that the Hamiltonian helps reduce the variance is still not convincing to me, given that there is no difference between the Hamiltonian raised in (7) and cumulative rewards. It would be more convincing if the authors could provide a more rigorous mathematical justification.

---

> > > > ### Author Response · Authors · 2022-08-09
> > > > **Response to Reviewer JucZ (1/4)**
> > > >
> > > > > First, in (2) the cumulative reward is the expectation of $Q^{\pi_\theta}$ with respect to the initial state distribution, which I believe is exactly the same as (7) except for the negative sign.
> > > >
> > > >
> > > > The authors realize that the equal sign in (7) may lead to some confusion and like to clarify it as follows.
> > > >
> > > > Please note that both the optimization objectives (2) and (7) of reinforcement learning are probabilistic functions, and the “Monte Carlo” method (Chapter 5 in [1]; [2]) is broadly used for gradient estimates whose operation involves a significant random component.
> > > >
> > > > * [1] Sutton, R. S., & Barto, A. G. (2018). Reinforcement learning: An introduction. MIT press.
> > > > * [2] Mohamed, S., Rosca, M., Figurnov, M., & Mnih, A. (2020). Monte Carlo Gradient Estimation in Machine Learning. J. Mach. Learn. Res., 21(132), 1-62.
> > > >
> > > > Therefore, the authors show that (2) and (7) are NOT exactly the same through the glass of Monte Carlo gradient estimators. This explanation is also available in Section 3.2 (the newly updated version) and will be added in the Appendix of the future version.
> > > >
> > > > To recap, inspired by [17], the authors formally reformulate (2) into a $K$-spin Hamiltonian equation
> > > >
> > > > $~~~~~~~~~~~~~~~~~~~~~~~~~~~~~~~~~~~~~~~~H(\theta) \triangleq -E_{S_0,A_0} [Q^{\pi_\theta}(S_0,A_0)],$
> > > >
> > > > $~~~~~~~~~~~~~~~~~~~~~~~~~~~~~~~~~~~~~~~~~~~~~~~~=-\lim_{K \rightarrow \infty}\sum_{k = 0}^{K-1} \sum_{\mu_0}^{\mathcal{S} \times \mathcal{A}} \cdots \sum_{\mu_k}^{\mathcal{S} \times \mathcal{A}} L_{\mu_0, ..., \mu_k} \pi_{\theta}(\mu_0)\cdots\pi_{\theta}(\mu_k),$
> > > >
> > > > $~~~~~~~~~~~~~~~~~~~~~~~~~~~~~~~~~~~~~~~~~~~~~~~~=- \lim_{K \rightarrow \infty} E_{\mu_0, \mu_1, ..., \mu_K} [\sum_{k = 0}^{K-1} L_{\mu_0, ..., \mu_k}]$,
> > > >
> > > > the expectation is taken over $S_0\sim d_0(\cdot),A_0\sim\pi_\theta(S_0,\cdot)$, and the density function $L_{\mu_0, ..., \mu_k}$ is given in (6).
> > > >
> > > > **Monte Carlo Estimator** [2]: Consider a general probabilistic objective $\mathcal{F}$ of the form:
> > > >
> > > > $~~~~~~~~~~~~~~~~~~~~~~~~~~~~~~~~~~~~~~~~~~~~~~~~~~~~~~~~~~~~~~~~\mathcal{F} \triangleq \mathbb{E}_{p(x;\theta)}[f(x;\phi)],$
> > > >
> > > > in which a function $f$ of an input variable $x$ with $\textit{structural parameters}$ $\phi$ is evaluated on average with respect to an input distribution $p(x; \theta)$ with $\textit{distributional parameters}$ $\theta$.
> > > >
> > > > A Monte Carlo method evaluates the function by first drawing independent samples $\hat{x}^{(1)},..., \hat{x}^{(N)}$ from the distribution $p(x; \theta)$, and then computing the average:
> > > >
> > > > $\widehat{\mathcal{F}}_N = \frac{1}{N} \sum_{i=1}^{N} f(\hat{x}^{(i)}), ~~where~~\hat{x}^{(i)} \sim p(x; \theta) ~for~i=1,...,N.$
> > > >
> > > > The Monte Carlo estimator for conventional objective (2) is
> > > >
> > > > $~~~~~~~~~~~~~~~~~~~~~~~~~~~~~~~~~~~~~~~~\widehat{J}(\theta) = \frac{1}{N}\sum\limits_{i=1}^{N} R(\tau^{(i)})$ where $\tau^{(i)} \sim P(\tau^{(i)} | \pi_{\theta})$ for $i=1,...,N,$
> > > > and
> > > > $~~~~~~~~~~~~~~~~~~~~~~~~~~~~~~~~~~~~~~~~P(\tau^{(i)} | \pi_{\theta}) = d_0(s_0^{(i)}) \cdot \prod_{k = 0}^{T} \mathbb{P}(s_{k + 1}^{(i)}| s_k^{(i)}, a_k^{(i)}) \pi_{\theta}(a_k^{(i)}|s_k^{(i)}).$
> > > >
> > > > The Monte Carlo estimator for the Hamiltonian reformulation is
> > > >
> > > > $~~~~~~~~~~~~~~~~~~~~~~~~~~~~~~~~~~~~~~~~\widehat{H}(\theta) = \frac{1}{N'}\sum\limits_{i=1}^{N'} \sum_{k = 0}^{K-1} L_{\mu_0^{(i)}, ..., \mu_k^{(i)}},$ for $i=1,...,N',$
> > > > and
> > > > $~~~~~~~~~~~~~~~~~~~~~~~~~~~~~~~~~~~~~~~~L_{\mu_0^{(i)}, ..., \mu_k^{(i)}} = \gamma^k\cdot R(\mu_k^{(i)})\cdot d_0(s_0^{(i)}) \cdot \prod_{\ell = 0}^{k - 1} \mathbb{P}(s_{\ell + 1}^{(i)}|\mu_{\ell}^{(i)}).$
> > > >
> > > > $\textbf{Remark:}$ The above two Monte Carlo estimators are quite different in the simulation process. The conventional Monte Carlo estimator samples a random trajectory by following an environment's stochastic transition and a policy. In contrast, our novel Hamiltonian Monte Carlo estimator measures a random path's discounted reward (the "energy") without following any policy, and the Hamiltonian equation combinatorially enumerates all possible paths of length $K$ over the state-action space. In other words, the simulation process of the Hamiltonian term does not rely on any policy. Therefore, the Hamiltonian term is a suitable regularizer for both on-policy and off-policy algorithms.
> > > >
> > > > This fundamental difference is due to the Ising model in (5), which combinatorially enumerates all paths and separates the environment and the policy.

---

> > > > > ### Author Response · Authors · 2022-08-09
> > > > > **Response to Reviewer JucZ (2/4)**
> > > > >
> > > > > > Second, PPO is an on-policy PG method that only solves the Bellman equation for the current policy. It does not solve the Bellman optimality equation.
> > > > >
> > > > > This is a great question and Reviewer Zi6K raises a very similar one. Therefore, the authors like to refer to the relevant discussions.
> > > > >
> > > > > > The main motivation is based on that the Bellman’s optimality equation, which is the base of Q-learning-like algorithms such as DDPG, has multiple fixed points. But the algorithm also work well on PPO, which even does not use Bellman equation to learn the value function. Can the authors provide any explanation about why it PPO+H works so well?
> > > > > >
> > > > > > > The authors believe the reviewer made a factual error comment that “PPO does not use Bellman equation to learn the value function”. PPO is a policy gradient algorithm with advantage function estimation.
> > > > > > >
> > > > > > > In the following reference [1], it is theoretically clear how a critic is plugged into the policy gradient theorem in equations (8) and (9). Thus, all actor-ciitic DRL algorithms use the Bellman equation to learn the value function.
> > > > > > >
> > > > > > > As mentioned in 216~218, the authors use GAE [28] for advantage estimation, where a value function is approximated as a baseline and optimized via the Bellman equation. The authors follow several benchmark implementations as in Stable Baseline3, RLlib, Tianshou, etc, which update the value function by minimizing the TD-residual. Therefore, it is reasonable that the H term also works with PPO.
> > > > > > >
> > > > > > > [1] Wen, Junfeng, et al. "Characterizing the gap between actor-critic and policy gradient." International Conference on Machine Learning. PMLR, 2021.
> > > > > > >
> > > > > > > > I appreciate the authors' reply to my questions.
> > > > > > > >
> > > > > > > > First, for PPO, what I meant was that PPO, which is an on-policy algorithm, does not use an Optimality Bellman Equation, i.e., no max_a operation in value learning. However, the multiple fixed points problem only occurs when the max_a operation is used like in Q-learning. This makes PPO failed to be an empirical evidence to support the authors' foundamental motivation (while I agree it is a good to see PPO+H works well). I also checked other reviewer's comments, which also raise question about the correlation between motivation (BE has multiple fixed points) and the empirical results.
> > > > > > > >
> > > > > > > > Second, for the analogy between policy and spin angular momentum. I am still confused about the analogy. Spin in quantum physics is either 1 or -1 when being measured. The authors said that non-optimal policy is the "orientation"--what I understand here is that the non-optimal policy is the spin vector's z-axis component on the bloch sphere, is this what the authors meant? If so, I do not see a intuition here to consider optimal policy as "being measuted oplicy". Could the authors explain in more details? I have a certain background in physics but I feel hard to fully grasp the intuition behind the analogy.
> > > > > > > >
> > > > > > > > Since most of readers of the NeurIPS conference come from computer science and computational neuroscience, and given the fact this work is not quantum RL but physics-inspired conventional deep RL, I think certain efforts need to be made to make the audience understand at least the intuitive motivation.
> > > > > > > >

---

> > > > > > ### Author Response · Authors · 2022-08-09
> > > > > > **Response to Reviewer JucZ (3/4)**
> > > > > >
> > > > > > > > > > Thanks very much for your clarification on your question “our motivation and empirical results” and “intuitive motivation”. It is indeed a good question and a good opportunity for the authors to emphasize the foundational contributions. Before replying, the authors made a thorough rechecking of Sutton’s RL book and survey of existing DRL algorithms.
> > > > > > > > > >
> > > > > > > > > > The authors would like to state several backgrounds of current deep reinforcement learning algorithms. First, there is a fact that the Optimality Bellman Equation (with a $\max_a$ operation) is an optimality condition, which originally is a sufficient condition, namely, any optimal policy of MDP and Dynamic Programming problems should satisfy the Optimality Bellman Equation. Note that it is not about algorithm design, however, several (deep) RL algorithms used it, like Q-learning, DQN, etc. Please do not use Q-learning and DQN algorithms as an example of how the Optimality Bellman Equation should be used.
> > > > > > > > > >
> > > > > > > > > > Here, the authors would like to point out that the optimality condition does not discuss how an algorithm should be designed or implemented, but a mathematical principle that any optimal policy should satisfy, as long as the target problem space possesses the MDP structure. Even if the $\max_a$operation is NOT used in an RL algorithm, an optimal policy should satisfy the Optimality Bellman Equation. However, such an optimal policy under the Optimality Bellman Equation is not unique. In practice, an algorithm will randomly converge to one of many policies. This foundational issue of Optimality Bellman Equation is our strong motivation to consider an alternative, Hamiltonian equation, which is universally used in modern physics.
> > > > > > > > > >
> > > > > > > > > > Second, the currently most widely used Actor-Critic algorithms (both DDPG and PPO) use Bellman equation (not the optimal one) for training the critic network (for value estimation), and the critic network converges when the Optimality Bellman Equation is satisfied. That is to say, since the Optimality Bellman Equation has multiple fixed points, the obtained critic network will randomly converge to one of many fixed points, thus the trained Actor-Critic agent also randomly converges to one of many fixed points.
> > > > > > > > > >
> > > > > > > > > > Third, Fig. 2 and Table. 2 (vanilla PPO) have given empirical verification about the observation that an DRL algorithm will randomly converge to one of many policies. Such an observation is widely recognized by the DRL community, there are YouTube videos (for example, an upside down policy of the HalfCheetah task: https://www.youtube.com/watch?v=qU8Nd9lyxlw). The authors believe that Fig. 1 (and descriptions in Introduction), Fig. 2, and Table. 2 together strongly motivate our work. Even though there are so many great works, the authors are happy to address such a foundational issue and present to the community an add-on term to mitigate the highly unstable performance of existing DRL algorithms, which is a major criticism from practitioners.
> > > > > > > > > > Fourth, differentiating “the training/learning process” and “the obtained optimal policy” is the key to understanding the novel analogy between MDP and K-spin Ising model. “Spin in quantum physics is either 1 or -1 when being measured”, similarly, an optimal policy assigns either 1 or 0 to each state-action pair, which is obtained when optimality is achieved. Please note that an optimal policy is deterministic, i.e., either 1 or 0 for each state-action pair. The authors believe that this fact may be the cause of the reviewer’s confusion. In other words, during the training process, a non-optimal policy is just like in a quantum superposition state; when the training process ends, an optimal policy is “measured” (when the algorithm is converged). Since both the initialization and the training process are random, it is natural to treat the training process as a quantum superposition state; and an optimal policy after convergence (and the Bellman’s optimality equation is satisfied) is just like being “measured” and becomes deterministic.
> > > > > > > > > >
> > > > > > > > > > Furthermore, both the Ising model and lowest-energy state are fundamental in physics. The authors are quite impressed by the fact that the Hamiltonian equation (simple and easy-to-implement) can be used as an add-on term to most actor-critic DRL algorithms (note that we tested over 5 algorithms, i.e., DDPG, PPO, SAC, TD3), and such an add-on term effectively addresses practitioners’ major criticism “unstable”. We are confident that this work will be highly recognized by both NeurIPS community members and industrial practitioners.

---

> > > > > > > ### Author Response · Authors · 2022-08-09
> > > > > > > **Response to Reviewer JucZ (4/4)**
> > > > > > >
> > > > > > > > Finally, the argument that the Hamiltonian helps reduce the variance is still not convincing to me, given that there is no difference between the Hamiltonian raised in (7) and cumulative rewards. It would be more convincing if the authors could provide a more rigorous mathematical justification.
> > > > > > >
> > > > > > > If the reviewer agrees with the above responses regarding H-term reformulation and Bellman’s optimality equation and agrees that the Hamiltonian equation in (7) is a novel foundational perspective, the authors would like to reason the impressive variance reduction results in Table 2 as follows:
> > > > > > > 1. There are multiple fixed points (policies) in existing DRL algorithms, due to Bellman’s Optimality Equation, as shown by three examples in Fig. 1 (the case of $\gamma = 1$) and Fig. 5 (the case of $\gamma \in (0, 1)$) which is further supported by the empirical experiments in Fig. 2.
> > > > > > > 1. The conventional PPO algorithm (note that we also tested other DRL algorithms) randomly converges to one of several policies, resulting in high variance. This is empirically shown in the third column of Table 2,
> > > > > > > 1. Ising Model in Table 1 measures the “energy” of a policy, thus the proposed $H$-term helps the policy network stably converge to a physically stationary policy, namely, the lowest-energy configuration of a system.
> > > > > > >
> > > > > > > Therefore, the Alg. 1 will converge to a policy with the smallest $H$-value, thus the variance of multiple trainings will be reduced.
> > > > > > >
> > > > > > > In Appx. E (newly updated version), we also provide a rigorous mathematical justification that the added H-term will result in a reduced variance of the gradient.

---

### Official Review · Reviewer_UzCH · 2022-07-07

**Rating:** 5
**Confidence:** 4
**Soundness:** 2 fair
**Presentation:** 3 good
**Contribution:** 3 good

**Summary:**

This work proposes to help a policy network stably find a stationary policy by making an analogy between an MDP and a quantum K-spin Ising model. To demonstrate the existence of multiple fixed points of the Bellman Optimality equation, the authors used three examples from dynamic programming. The paper empirically evaluated the performance of the newly proposed method on 6 MuJoCo tasks.

**Questions:**

1. Rather than an analogy between optimal policy and quantum field, should it be just policy with the quantum field?
2. How much more computational cost is needed for the additional H term as compared to the baseline methods?
3. if the main point is to make an analogy with physical systems and MDP, why the quantum k-spin Ising model is specifically chosen?
4. In the Broader Impace Statement the authors state that 'bring together the strengths of both approaches and yield new insights in both fields'. However, I'm not sure what this can bring for the quantum community?

**Limitations:**

1. Experiment is limited to the 6 MuJoCo tasks.
2. The analogy to 'lowest energy' makes me worry that this method only works for physical tasks (humanoid, hopper, etc.).
3. Lacking theoretical analysis on the resulting new algo.
4. Maybe can include a comparison of the computational cost of this method versus trainiing ten times (depends on how many would be required to reach the same level of stability) of the original DRL agent.

**Strengths And Weaknesses:**

Strengths:
Interesting problem and approach. The instability of DRL algorithms is definitely a major concern.
Derivation seems to be sound.

Weaknesses:
The experiment is quite limited (6 MuJoCo tasks)
Number of compared baseline is too few & did not compare the performance with any quantum RL/Hamiltonian mechanics method
The computational cost may prohibit its ability to scale to more complex problems

---

> ### Author Response · Authors · 2022-08-02
> **Response to Reviewer UzCH**
>
> Thank you for your insightful feedback. We would like to address your concerns and answer your questions in the following.
>
> > Rather than an analogy between optimal policy and quantum field, should it be just policy with the quantum field?
>
> Both optimal policy $\pi^* \in$ {-1, 1} and policy $\pi \in [0, 1]$ could be naturally mapped to a quantum field (a spin configuration). We agree that there was notation reuse and we did not make it explicit. In physics, a spin orients at an angle $\in [0, 2\pi)$ and takes continuous value $\in [-1, 1]$, while the optimal spin configuration takes discrete value {-1, 1}. Therefore, the optimal policy $\pi^*$ corresponds to the optimal spin configuration, and the policy π corresponds to the case when spins take continuous values. The authors add a new row in Table 1 to help distinguish the mapping for optimal policy $\pi^*$ and policy $\pi$.
>
>
> > How much more computational cost is needed for the additional H term as compared to the baseline methods?
>
> There is a relatively little computational cost when the Hamiltonian gradient is truncated with a small K, say K=24 in Table 2. The authors provided a complexity analysis in Section 4.2 lines 201~205, in which the additional cost only involves a Hamiltonian gradient computation.
>
> For the reviewer’s concern about the computational cost, as mentioned in lines 263 and 295~296, the authors foresee potential high computational costs for future works if Bellman equations in RL would be replaced by the Hamiltonian equation. Note that the accuracy of the H term approximation is directly related to the K-truncation. Therefore, future works may require a very accurate estimate and need a larger K, which may experience high computational costs.
>
>
> > if the main point is to make an analogy with physical systems and MDP, why the quantum k-spin Ising model is specifically chosen?
>
> There are two main motivations behind it. The K-spin Ising model matches the sequential decision-making process and the Hamiltonian equation measures the energy of an Ising model (here our policy network).
>
> On the other hand, the Ising model is a universal model, e.g., NP-hard problems [1], and iterative optimization algorithms [2].
>
> * [1] Lucas, Andrew. "Ising formulations of many NP problems." Frontiers in physics (2014): 5.
> * [2] Li, Ke, and Jitendra Malik. "Learning to Optimize." ICLR, 2017.
>
>
>
> > In the Broader Impace Statement the authors state that 'bring together the strengths of both approaches and yield new insights in both fields'. However, I'm not sure what this can bring for the quantum community?
>
> There are two aspects that our work will bring insights to the quantum community.
>
> First of all, our work is trying to bring the success of DRL algorithms to the quantum RL field, which is an active research area in the quantum machine learning community.
>
> On the other hand, RL has been an alternative promising approach for solving quantum physics problems, such as CQ, QC, and QQ problems, depending on whether the agent (first symbol) or the environment (second symbol) are classical (C) or quantum (Q).  One recent breakthrough is using RL to control nuclear fusion [1].
>
> Moreover, the ML community is also very interested in borrowing quantum mechanisms for two major reasons. One is that quantum mechanisms may deliver quadratic improvements in learning efficiency and exponential improvements in performance over limited time periods [2, 3].
>
> * [1] Degrave, Jonas, et al. "Magnetic control of tokamak plasmas through deep reinforcement learning." Nature 602.7897 (2022): 414-419.
> * [2] Biamonte, Jacob, et al. "Quantum machine learning." Nature 549.7671 (2017): 195-202.
> * [3] Dunjko, Vedran, Jacob M. Taylor, and Hans J. Briegel. "Quantum-enhanced machine learning." Physical review letters 117.13 (2016): 130501.

---

### Official Review · Reviewer_65t2 · 2022-07-10

**Rating:** 3
**Confidence:** 3
**Soundness:** 2 fair
**Presentation:** 1 poor
**Contribution:** 2 fair

**Summary:**

The paper posits that the Bellman optimality operator has multiple fixed-points. It becomes apparent that in defining a discounted MDP, the paper allows discount factors $\gamma = 1$ in the infinite-horizon setting contrary to conventional wisdom in RL. Arguing that the existence of multiple fixed-points is a key source of high variance in RL, the work draws inspiration from statistical mechanics to regularize actor-critic and policy gradient algorithms, and presents results over PPO and DDPG with reduced variance across seeds and improved average performance.

**Questions:**

- In L73, the discount factor is defined as $\gamma \in (0, 1]$. In L127, $\gamma \in (0, 1)$. Why the difference?
- L114: there is no summation in (6), so why is this called a _cumulative_ reward?
- In (7), what does a summation from $\mu_k$ to $\mathcal{S} \times \mathcal{A}$ mean?
- L265: if memory budget permits replay buffer size 800 for K=24, for a fair comparison it would make sense to set the buffer size to 800 for K=8 and K=16 too.
- $R(\tau)$ should be defined after (2).

**Limitations:**

As far as I can tell, there is no discussion of limitations, except maybe the complexity trade-off due to the truncation parameter K. I'm curious as to whether there exist MDPs such that Hamiltonian regularization negatively impacts performance.

**Strengths And Weaknesses:**

Weaknesses:

- In both theory and practice, practitioners typically set $\gamma < 1$ in infinite horizon settings, in which case the Bellman optimality operator has a unique fixed point due to the monotonicity and contraction properties, and the Banach fixed-point theorem. That being said, there exist some exceptional cases such as those discussed in Ch. 3 of the Bertsekas ADP textbook, where discounting with $\gamma < 1$ may fail to find the optimum policy and additional restrictions on the space of value functions is necessary (Bertsekas, 2019). However, I think saying that the Bellman optimality operator has multiple fixed points (in bold and italics, multiple times) without making it very clear early on that the setting involves $\gamma \in [0, 1]$ rather than $\gamma \in [0, 1)$ is misleading. Furthermore, there are lots of other valid sources of variance in deep RL including but not limited to optimization of non-convex/non-stationary objectives, stochastic gradients, reward sparsity, initial conditions, complexity of learner function class, etc. that the wording in this paper neglects. I don't suppose existence of multiple fixed points is a primary concern since practitioners use $\gamma < 1$ in infinite-horizon (and long-horizon) settings. So, I fail to see the motivation behind the proposed regularization approach and remain deeply skeptical of the soundness of this paper.
- I found the paper to be rather difficult to read. The paper could use copy editing.
- The paper claims to fix the multiple fixed-point issue with the Hamiltonian regularization scheme, but only shows the effect of its usage for a few pedagogical examples. But I would expect to see a theorem attached to this claim with a mathematical proof of how the issue is fixed under the proposed framework.

Strengths:
- Despite the awkwardness of the motivation, story and positioning of the paper, I can see some merit in regularizing the policy in the way that this paper proposes. Indeed, temporal regularization has been shown to serve as a good variance reduction technique when applied to the value function [33]. While [8] proposes a form of temporal regularization on the policy with a control prior, their approach requires having access to some known dynamics, while this paper does not. So, I think a major revision of the paper with better motivation, presentation and discussion of related work has potential.

---

> ### Author Response · Authors · 2022-08-02
> **Response to Reviewer 65t2 (1/2)**
>
> Thank you for your feedback. We would like to address your concerns and your questions in the following.
>
> > In both theory and practice, practitioners typically set $\gamma < 1$ in infinite horizon settings, in which case the Bellman optimality operator has a unique fixed point due to the monotonicity and contraction properties, and the Banach fixed-point theorem. That being said, there exist some exceptional cases such as those discussed in Ch. 3 of the Bertsekas ADP textbook, where discounting with $\gamma < 1$ may fail to find the optimum policy and additional restrictions on the space of value functions is necessary (Bertsekas, 2019). However, I think saying that the Bellman optimality operator has multiple fixed points (in bold and italics, multiple times) without making it very clear early on that the setting involves $\gamma \in [0, 1]$ rather than $\gamma \in [0, 1)$ is misleading.
>
> The reviewer’s objection seems to be highly relying on his/her misreading that our results ONLY hold for the undiscounted case $\gamma=1$. The authors believe there are several factual errors regarding “motivation”, “soundness” and “practical usefulness”, resulting in highly biased comments on this paper.
>
> First, the authors discussed the case of $\gamma=1$ and the case of $\gamma \in (0,1)$ separately.
> * In the Introduction (from line 21 to line 41), for easy understanding, the authors described the Bellman equation’s issue of multiple fixed points with three motivating examples of $\gamma=1$.
> * The authors deferred the more complex case of $\gamma \in (0,1)$ and pointed out that “more examples are given in Fig. 5 and Appx. A”. For some unknown reason, the reviewer ignored the continued discussion. In the revised version, the authors change the wording from “more examples” to “examples with $\gamma < 1$”.
>
> > Furthermore, there are lots of other valid sources of variance in deep RL including but not limited to optimization of non-convex/non-stationary objectives, stochastic gradients, reward sparsity, initial conditions, complexity of learner function class, etc. that the wording in this paper neglects. I don't suppose existence of multiple fixed points is a primary concern since practitioners use $\gamma < 1$ in infinite-horizon (and long-horizon) settings. So, I fail to see the motivation behind the proposed regularization approach and remain deeply skeptical of the soundness of this paper.
>
> Second, the existence of multiple policies is also common in practical tasks with $\gamma < 1$, as mentioned in recent studies (benchmarks) [1, 2, 3], which contradicts the reviewer’s comment “I don't suppose existence of multiple fixed points is a primary concern since practitioners use $\gamma < 1$ in infinite-horizon (and long-horizon) settings”. The observational experiments on MuJoCo tasks in Section 2.2 further verify the issue, and the experiments in Section 5 demonstrate the practical usefulness of the H term, where MuJoCo tasks are standard benchmarking tasks for continuous control.
>
> Third, the authors believe that this paper targets the issue of multiple fixed points, while other sources like “optimization of non-convex/non-stationary objectives, stochastic gradients, reward sparsity, initial conditions, complexity of learner function class” are out of the scope. For completeness, we summarize the sources in the Introduction of the revised version.
> *  [1] Duan, Yan, et al. "Benchmarking deep reinforcement learning for continuous control." International Conference on Machine Learning. PMLR, 2016.
> *  [2] Eysenbach, Benjamin, et al. "Diversity is all you need: Learning skills without a reward function." International Conference on Learning Representations. 2018.
> *  [3] Recht, Benjamin. "A tour of reinforcement learning: The view from continuous control." Annual Review of Control, Robotics, and Autonomous Systems 2 (2019): 253-279.

---

> > ### Author Response · Authors · 2022-08-02
> > **Response to Reviewer 65t2 (2/2)**
> >
> > > I found the paper to be rather difficult to read. The paper could use copy editing.
> >
> > The authors will update the manuscript to improve the readability.
> >
> > > The paper claims to fix the multiple fixed-point issue with the Hamiltonian regularization scheme, but only shows the effect of its usage for a few pedagogical examples. But I would expect to see a theorem attached to this claim with a mathematical proof of how the issue is fixed under the proposed framework.
> >
> > In addition to showing the effectiveness of the H term on three examples in both undiscounted and discounted cases, the authors provide experimental results (with visualization results in supplementary materials) on six MuJoCo tasks. Due to the high-dimensional continuous state and action space, these tasks are widely-recognized challenging tasks in robotic control [1].
> >
> > The current work is not theoretical. It provides a physical-inspired algorithm design that is easy to implement, delivering significant improvements in performance. The target issue of instability of DRL algorithms is practically important for RL’s adoption in real-world tasks, say robotic control.
> >
> >
> > > In L73, the discount factor is defined as $\gamma \in (0, 1]$. In L127, $\gamma \in (0, 1)$. Why the difference?
> >
> > Here we are discussing a practical case of discounted cumulative rewards. $\gamma < 1$ is required to guarantee a small approximation error.
> >
> >
> > > L114: there is no summation in (6), so why is this called a cumulative reward?
> >
> > Thanks for the careful reading, and the typo is fixed in the revised version.
> >
> > > In (7), what does a summation from $\mu_k$ to $\mathcal{S} \times \mathcal{A}$ mean?
> >
> > The summation comes from the standard Hamiltonian equation as defined in (5).
> >
> >
> > > L265: if memory budget permits replay buffer size 800 for K=24, for a fair comparison it would make sense to set the buffer size to 800 for K=8 and K=16 too.
> >
> > Thanks for the suggestion, and the authors will provide an experiment using the buffer size 800 for all K values in the Appendix G.2 of the revised version.
> >
> > > $R(\tau)$ should be defined after (2).
> >
> > Yes, it should be given right after (2). The cumulative reward along a trajectory $\tau$.

---

> > ### Comment · Reviewer_65t2 · 2022-08-08
> > **On the supposed issue of multiple fixed points**
> >
> > Upon reading the authors' response to all the reviews including mine, it seems to me that the paper suffers from a major confusion. The Bellman operators operate on the space of bounded real-valued functions over $\mathcal{S}$ or $\mathcal{S} \times \mathcal{A}$, **not the space of policies**. Therefore, the fixed point of the Bellman equations (or the Bellman operators) are not policies, but value functions. Said value functions are guaranteed to be unique whenever $\gamma \in [0, 1)$. However, multiple distinct policies may have the same optimal value function, e.g., when $Q^*(\mathbf{s}, \mathbf{a}_1) = Q^*(\mathbf{s}, \mathbf{a}_2)$, we may have $\pi_1(\mathbf{s}) = \mathbf{a}_1$ and $\pi_2(\mathbf{s}) = \mathbf{a}_2$, yet $V^{\pi_1}(\mathbf{s}) = V^{\pi_2}(\mathbf{s}) = V^*(\mathbf{s})$. Seeing as the paper confuses the uniqueness of the value function (which is the fixed point of the Bellman operator) with the uniqueness of the policy, and this confusion is highlighted *10 times* throughout the paper, I think that the paper needs a major revision, so I am keeping my score.
> >
> > I also *disagree* with the authors' argument that the paper is of an empirical nature and therefore a proof of how the variance issue is addressed by Hamiltonian regularization is not needed. I think the paper makes big claims about (i) the high variance issue in RL arising from the non-uniqueness of the policy corresponding to the Bellman fixed-points, (ii) addressing the Bellman issue and consequently the variance issue. However, the empirical evidence provided is rather indirect and not sufficient to convince a wide audience of which I am a member.

---

> > > ### Author Response · Authors · 2022-08-09
> > > **About Multiple Policies and Variance Reduction by the Hamiltonian Regularization (1/2)**
> > >
> > > Thanks very much for this detailed clarification of possible confusion. The authors would like to take this opportunity to communicate with the reviewer.
> > >
> > > First, the authors agree that “the Bellman operators operate on the space of bounded real-valued functions over $S$ or $S\times A$, not the space of policies.” In Fig. 1 (the case of $\gamma = 1$) and Fig. 5 (the case of $\gamma \in (0, 1)$), we show that there are multiple feasible solutions for the value function, not limiting to the reviewer’s mentioned case “the same optimal value function, e.g., when $Q^*(s,a1)=Q^*(s,a2)$, or $V^\pi_1(s)=V^\pi_2(s)=V^*(s)$”,  but different functions $Q$’s or $V$’s.
> > >
> > > Second, the authors are quite aware of the well-known uniqueness result that  “the Bellman optimality operator has a unique fixed point due to the monotonicity and contraction properties, and the Banach fixed-point theorem”. For example, the Bellman optimality operator is a contraction over a complete metric space of real numbers with a metric L-infinity norm. The authors would like to point out that such a result holds under certain sufficient conditions.
> > >
> > > Third, the authors would like to list the following evidence that there are multiple policies,
> > > 1. In the three examples of the case of $\gamma \in (0, 1)$ in Fig. 5, there are multiple policies with different value functions. More examples can be found in Ch. 3.1 of the Bertsekas ADP textbook [1] (http://web.mit.edu/dimitrib/www/AbstractDP_ED3_TEXT_2021.pdf)
> > > 1. Section 3 Counterexamples and Section 3.1 Multiple Fixed Points of [2] give counter-examples of the uniqueness solution and also examples for multiple fixed points. Also, [3] pointed out that “However, it is not then possible to assure uniqueness of the fixed point on $C(X)$. Also, in this case, a convergence of the successive approximations from an arbitrary element of $C(X)$ can fail.”
> > > 1.  For the Six MuJoCo tasks in Fig. 2 and Table 2 and more tasks in [4][5][6], there is empirical evidence of “a trained agent randomly converges to one of the multiple policies”.
> > > 1.  Besides the above robotic control tasks, the authors observed the multiple policies issue by checking the MDP instances of several NP-hard problems, e.g., Graph MaxCut, Minimum set cover, Mixed integer programming problems (MILP). Actually, the issue of multiple policies is currently a major challenge for DRL solutions that do not always beat commercial solvers (Gurobi and SCIP).
> > >
> > > Even though there are so many great works, the authors are happy to address such a foundational issue and present to the community an add-on term to mitigate the highly unstable performance of existing DRL algorithms, which is a major criticism from practitioners.
> > >
> > >
> > > * [1] Bertsekas D. Abstract dynamic programming[M]. Athena Scientific, 2022.
> > > * [2] Kamihigashi, T. (2012). Existence and uniqueness of a fixed point for the Bellman operator in deterministic dynamic programming (No. DP2012-05).
> > > * [3] Rincón‐Zapatero, Juan Pablo, and Carlos Rodríguez‐Palmero. "Existence and uniqueness of solutions to the Bellman equation in the unbounded case." Econometrica 71.5 (2003): 1519-1555.
> > > * [4] Duan, Yan, et al. "Benchmarking deep reinforcement learning for continuous control." International Conference on Machine Learning. PMLR, 2016.
> > > * [5] Eysenbach, Benjamin, et al. "Diversity is all you need: Learning skills without a reward function." International Conference on Learning Representations. 2018.
> > > * [6] Recht, Benjamin. "A tour of reinforcement learning: The view from continuous control." Annual Review of Control, Robotics, and Autonomous Systems 2 (2019): 253-279.

---

> > > > ### Author Response · Authors · 2022-08-09
> > > > **About Multiple Policies and Variance Reduction by the Hamiltonian Regularization (2/2)**
> > > >
> > > > Next, we would like to provide our response to “the reviewer expects to see a theorem attached to this claim with a mathematical proof of how the issue is fixed under the proposed framework” and “a proof of how the variance issue is addressed by Hamiltonian regularization is needed.”
> > > >
> > > > In Appx. E (newly updated version), we also provide a rigorous mathematical justification that the added H-term will result in a reduced variance of the gradient. And the authors would like to reason the impressive variance reduction results in Table 2 as follows:
> > > > 1. There are multiple fixed points (policies) in existing DRL algorithms, due to Bellman’s Optimality Equation, as shown by three examples in Fig. 1 (the case of $\gamma = 1$) and Fig. 5 (the case of $\gamma \in (0, 1)$) which is further supported by the empirical experiments in Fig. 2.
> > > > 1. The conventional PPO algorithm (note that we also tested other DRL algorithms) randomly converges to one of several policies, resulting in high variance. This is empirically shown in the third column of Table 2,
> > > > 1. Ising Model in Table 1 measures the “energy” of a policy, thus the proposed H-term helps the policy network stably converge to a physically stationary policy, namely, the lowest-energy configuration of a system.
> > > > 1. Therefore, the Alg. 1 will converge to a policy with the smallest $H$-value, thus the variance of multiple trainings will be reduced.

---

### Official Review · Reviewer_Zi6K · 2022-07-10

**Rating:** 7
**Confidence:** 4
**Soundness:** 3 good
**Presentation:** 2 fair
**Contribution:** 2 fair

**Summary:**

The paper first suggests that the Bellman's optimality equation has multiple fixed points. Then a analogy is made between MDP and qunatum K-spin Ising model, and a reformulation of expected return into quantum K-spin Hamiltonian equation is proposed. It is argued that by regularizing the policy to have a stationary Hamiltonian,  the model can 1). achieves a relative high reward independent of the initialization; and 2). is robust to interference/noise in the inference stage, and thus reduce performance variance among random seeds. This idea has been practically implemented by randomly sampling consecutive trajectories from a specific replay buffer and minimizing the Hamiltonian by policy gradient. The experiments on MuJoCo robotic control tasks have shown the effectiveness of the proposed methods using both DDPG and PPO as base algorithms, in terms of slightly higher mean performance and significantly lower variance. Furthermore, the agents converged to the stationary policy with a substantially higher ratio with the proposed method.

**Questions:**

[Major]
- Line 48: "we make a novel analogy between an MDP and a quantum K-spin Ising model". However, ref [21] proposed to model MDP with K-spin Hamiltonian. What is the difference?
- The optimal policy function $\pi^*$ and a general policy $\pi$ are mixed-up. E.g., line 111,  I understand the optimal policy $\pi^*(\mu_k) \in  $ {0 ,1}   can be mapped to spin operator, which is a common practice in quantum computation. However, how about $\pi(\mu_k)$, which is a continuous-value scalar? Also, in Table 1, the optimal policy is analogous to the spin operators, while the Hamiltonian the functional of non-optimal policy. I am a bit confused about this mixing-up.
- The main motivation is based on that the Bellman’s optimality equation, which is the base of Q-learning-like algorithms such as DDPG, has multiple fixed points. But the algorithm also work well on PPO, which even does not use Bellman equation to learn the value function. Can the authors provide any explanation about why it PPO+H works so well?
- There is a lack of discussion about related work in deep RL to reduce variance, e.g., https://openreview.net/pdf?id=9xhgmsNVHu
- My personal opinion is that the analogy is confusing to people without quantum physics background. And in the end, the algorithm used the classic policy gradient algorithm to estimate the Nabla of Hamiltonian. I feel that Algorithm 1 itself is indeed intuitive (heuristic) without introducing quantum physics. My suggestion is to defer some details of the analogy to the appendix (then the authors can have more space  to clearly explain it), and complement the main texts with contents such as related work and ablation studies.


[Minor]
- Line 47: I suggest to add "often" or "sometimes" because there are cases we want some diversity, e.g., option-critic and DIAYN.
- Line 87: polices --> policies
- Fig.4 is not centered.


**Strengths And Weaknesses:**

[Strength]
1. The paper touches a relatively important problem in deep RL, namely how to reduce variance of policies with different random seeds.
2. The proposed method is simple and easy to implement.
3. The experimental results are good, which show the effectiveness of the proposed methods by reducing variance by 65.2% ~ 85.6%
4. 3 simple yet motivated examples to show that the Bellman’s optimality equation has multiple fixed points.

[Weakness]
1. Lack of discussion of related work.
2. The analogy is hard for a reader without quantum physics background.
3. Some of the paper's claims need to be further supported.
4. The paper sometimes mixes-up the optimal policy function $\pi^*$ and non-optimal policy $\pi$, making the analogy a bit confusing.

See below for the details of my concerns.

---------------------- post-rebuttal -----------------
The author has resolved most of my conerns and agreed to update the manuscript to address the issues. Correspondingly, I update my score toward acceptance, mainly because the results (Table 2) are appealing with a relatively simple add-on (H term). Nonetheless, I expect the authors in the future to more comprehensively investiagte the motivation using empirical and theoretical analysis to support their claims.

---

> ### Author Response · Authors · 2022-08-02
> **Response to Reviewer Zi6K (1/3)**
>
> Thank you for your thoughtful comments. We would like to address your concerns and your questions in the following.
>
> >Line 48: "we make a novel analogy between an MDP and a quantum K-spin Ising model". However, ref [21] proposed to model MDP with K-spin Hamiltonian. What is the difference?
>
> Ref [21] modeled MDP with K-spin Hamiltonian (in Section III), and made an analogy between an MDP and classic field theory in Table I.
> There are several major differences:
> 1. The objective function (10) in [21] has a penalty term (for their quantum optimization approach), while in our deep reinforcement learning (DRL) approach, it is automatically satisfied by employing a softmax function. Moreover, (10) in [21] is the objective function of a quantum optimization task, while we used the Hamiltonian equation as an add-on term (a regularizer) for existing actor-critic DRL algorithms.
> 2. Their quantum optimization approach relies on the variational optimality condition (analogy to the Bellman optimality in DRL) and is amenable to quantum simulated annealing algorithms. Here, we use the K-spin Hamiltonian equation to regularize the policy network. A new policy gradient is added in Alg. 1 (line 15).
> 3. Their solution discussed the potential implementation on rear-term quantum hardware. Here, our major conclusion is that K-spin Hamiltonian can help reduce the high variance of DRL algorithms, which is brought by the Bellman equation’s issue of multiple fixed points.
> 4. Actually, the analogy (in ref [21]) between an MDP and classic field theory in Table I is not physically right. One should replace the classic field by a transverse field (a quantum field). First, there are no corresponding concepts of classic field’s potential energy and kinetic energy in RL. Second, the most important “conservation law” of classic field theory does not have a counterpart in RL. In contrast, our analogy to a quantum K-spin Ising model is more accurate, since the K-spin Ising model matches the sequential decision-making process, and the Hamiltonian equation measures the energy of an Ising model (here our policy network).
>
> > The optimal policy function $\pi^*$ and a general policy $\pi$ are mixed-up. E.g., line 111, I understand the optimal policy $\pi^* \in$ {0, 1} can be mapped to spin operator, which is a common practice in quantum computation. However, how about $\pi(\mu_k)$, which is a continuous-value scalar? Also, in Table 1, the optimal policy is analogous to the spin operators, while the Hamiltonian the functional of non-optimal policy. I am a bit confused about this mixing-up.
>
> We agree that there was notation reuse and we did not make it explicit on purpose. However, this is a quite standard routine in both algorithmic design and theoretical analysis. In physics, a spin orients at an angle $\in [0, 2\pi)$ and takes continuous value $\in [-1, 1]$, while the optimal spin configuration takes discrete value $\in$ {-1, 1}. Therefore, both optimal policy $\pi^* \in$ {-1, 1} and policy $\pi \in [0, 1]$ could be naturally mapped to a quantum field (a spin configuration), through the mapping in lines 111~114. Physicists study the simplified case with a spin $\in$ {-1, 1} since it already delivers theoretical results of phase transitions; in physical experiments, a spin takes values $\in [-1, 1]$.
>
> Revision: in Table 1, the authors add a new row to further clarify it. Note that the authors follow the routine in physics that the configuration can take either continuous values $\in [-1, 1]$ or discrete values $\in$ {-1, 1}, depending on the context.
> Some exemplar references, where the spin angle takes value $\in [0, \pi/2]$.
> * [1] Stoudenmire, Edwin, and David J. Schwab. "Supervised learning with tensor networks." Advances in Neural Information Processing Systems 29 (2016).
> * [2] Huggins, W., Patil, P., Mitchell, B., Whaley, K. B., & Stoudenmire, E. M. (2019). Towards quantum machine learning with tensor networks. Quantum Science and technology, 4(2), 024001.

---

> > ### Author Response · Authors · 2022-08-02
> > **Response to Reviewer Zi6K (2/3)**
> >
> > > The main motivation is based on that the Bellman’s optimality equation, which is the base of Q-learning-like algorithms such as DDPG, has multiple fixed points. But the algorithm also work well on PPO, which even does not use Bellman equation to learn the value function. Can the authors provide any explanation about why it PPO+H works so well?
> >
> > The authors believe the reviewer made a factual error comment that “PPO does not use Bellman equation to learn the value function”. PPO is a policy gradient algorithm with advantage function estimation.
> >
> > In the following reference [1], it is theoretically clear how a critic is plugged into the policy gradient theorem in equations (8) and (9). Thus, all actor-ciitic DRL algorithms use the Bellman equation to learn the value function.
> >
> > As mentioned in 216~218, the authors use GAE [28] for advantage estimation, where a value function is approximated as a baseline and optimized via the Bellman equation. The authors follow several benchmark implementations as in Stable Baseline3, RLlib, Tianshou, etc, which update the value function by minimizing the TD-residual. Therefore, it is reasonable that the H term also works with PPO.
> >
> > * [1] Wen, Junfeng, et al. "Characterizing the gap between actor-critic and policy gradient." International Conference on Machine Learning. PMLR, 2021.

---

> > > ### Author Response · Authors · 2022-08-02
> > > **Response to Reviewer Zi6K (3/3)**
> > >
> > > > There is a lack of discussion about related work in deep RL to reduce variance, e.g., https://openreview.net/pdf?id=9xhgmsNVHu
> > >
> > > The authors acknowledge missing this closely related work, as that paper was presented two weeks before the NeurIPS’ submission deadline. After a careful review, the authors will add more relevant works, including the above one suggested by the reviewer:
> > > * [2] Bjorck, Johan, Carla P. Gomes, and Kilian Q. Weinberger. "Is High Variance Unavoidable in RL? A Case Study in Continuous Control." International Conference on Learning Representations. 2021.
> > > * [3] Islam, Riashat, et al. "Reproducibility of Benchmarked Deep Reinforcement Learning Tasks for Continuous Control." RML workshop, ICML,  2017.
> > > * [4] Nikishin, Evgenii, et al. "Improving stability in deep reinforcement learning with weight averaging." Uncertainty in artificial intelligence workshop on uncertainty in Deep learning. 2018.
> > >
> > > > My personal opinion is that the analogy is confusing to people without quantum physics background. And in the end, the algorithm used the classic policy gradient algorithm to estimate the Nabla of Hamiltonian. I feel that Algorithm 1 itself is indeed intuitive (heuristic) without introducing quantum physics. My suggestion is to defer some details of the analogy to the appendix (then the authors can have more space to clearly explain it), and complement the main texts with contents such as related work and ablation studies.
> > >
> > > The authors foresee the reading difficulties for scholars without a quantum physics background, thus providing Table 1 and lines 111~119 for a detailed presentation. The authors like to point out that the H term is a physically inspired algorithm and therefore believe it is necessary to provide the novel analogy in the main body. Given the universality of the quantum K-spin Ising model, it is reasonable that the derived formula is intuitive.
> > >
> > > The authors have to say that ablation studies are naturally included.  1). H-term is an add-on term, so we compare algorithms with and without it; 2). For DDPG, we also compare with DDPG+PER for fairness; and 3). For the key parameter K (steps), we show the results for K=8, K=16, and K=24 in Table 2.
> > >
> > > > Line 47: I suggest to add "often" or "sometimes" because there are cases we want some diversity, e.g., option-critic and DIAYN.
> > >
> > > > Line 87: polices --> policies
> > >
> > > > Fig.4 is not centered.
> > >
> > > The authors thank the reviewer for the careful reading, and typos are fixed in the revised version.

---

> > > > ### Comment · Reviewer_Zi6K · 2022-08-04
> > > > **Thanks for the reply**
> > > >
> > > > I appreciate the authors' reply to my questions.
> > > >
> > > > First, for PPO, what I meant was that PPO, which is an on-policy algorithm, does not use an Optimality Bellman Equation, i.e., no max_a operation in value learning. However, the multiple fixed points problem only occurs when the max_a operation is used like in Q-learning. This makes PPO failed to be an empirical evidence to support the authors' foundamental motivation (while I agree it is a good to see PPO+H works well). I also checked other reviewer's comments, which also raise question about the correlation between motivation (BE has multiple fixed points) and the empirical results.
> > > >
> > > > Second, for the analogy between policy and spin angular momentum. I am still confused about the analogy.  Spin in quantum physics is either 1 or -1 when being measured. The authors said that non-optimal policy is the "orientation"--what I understand here is that the non-optimal policy is the spin vector's z-axis component on the bloch sphere, is this what the authors meant? If so, I do not see a intuition here to consider optimal policy as "being measuted oplicy". Could the authors explain in more details? I have a certain background in physics but I feel hard to fully grasp the intuition behind the analogy.
> > > >
> > > > Since most of readers of the NeurIPS conference come from computer science and computational neuroscience, and given the fact this work is not quantum RL but physics-inspired conventional deep RL, I think certain efforts need to be made to make the audience understand at least the intuitive motivation.

---

> > > > > ### Author Response · Authors · 2022-08-07
> > > > > **Response to Reviewer Zi6K (cont.)**
> > > > >
> > > > > Thanks very much for your clarification on your question “our motivation and empirical results” and “intuitive motivation”. It is indeed a good question and a good opportunity for the authors to emphasize the foundational contributions. Before replying, the authors made a thorough rechecking of Sutton’s RL book and survey of existing DRL algorithms.
> > > > >
> > > > > The authors would like to state several backgrounds of current deep reinforcement learning algorithms.
> > > > > First, there is a fact that the Optimality Bellman Equation (with a $\max_a$ operation) is an optimality condition, which originally is a sufficient condition, namely, any optimal policy of MDP and Dynamic Programming problems should satisfy the Optimality Bellman Equation. Note that it is not about algorithm design, however, several (deep) RL algorithms used it, like Q-learning, DQN, etc. Please do not use Q-learning and DQN algorithms as an example of how the Optimality Bellman Equation should be used.
> > > > >
> > > > > Here, the authors would like to point out that the optimality condition does not discuss how an algorithm should be designed or implemented, but a mathematical principle that any optimal policy should satisfy, as long as the target problem space possesses the MDP structure. Even if the $\max_a$operation is NOT used in an RL algorithm, an optimal policy should satisfy the Optimality Bellman Equation. However, such an optimal policy under the Optimality Bellman Equation is not unique. In practice, an algorithm will randomly converge to one of many policies. This foundational issue of Optimality Bellman Equation is our strong motivation to consider an alternative, Hamiltonian equation, which is universally used in modern physics.
> > > > >
> > > > > Second, the currently most widely used Actor-Critic algorithms (both DDPG and PPO) use Bellman equation (not the optimal one) for training the critic network (for value estimation), and the critic network converges when the Optimality Bellman Equation is satisfied. That is to say, since the Optimality Bellman Equation has multiple fixed points, the obtained critic network will randomly converge to one of many fixed points, thus the trained Actor-Critic agent also randomly converges to one of many fixed points.
> > > > >
> > > > > Third, Fig. 2 and Table. 2 (vanilla PPO) have given empirical verification about the observation that an DRL algorithm will randomly converge to one of many policies. Such an observation is widely recognized by the DRL community, there are YouTube videos (for example, an upside down policy of the HalfCheetah task: https://www.youtube.com/watch?v=qU8Nd9lyxlw). The authors believe that Fig. 1 (and descriptions in Introduction), Fig. 2, and Table. 2 together strongly motivate our work. Even though there are so many great works, the authors are happy to address such a foundational issue and present to the community an add-on term to mitigate the highly unstable performance of existing DRL algorithms, which is a major criticism from practitioners.
> > > > > Fourth, differentiating “the training/learning process” and “the obtained optimal policy” is the key to understanding the novel analogy between MDP and K-spin Ising model. “Spin in quantum physics is either 1 or -1 when being measured”, similarly, an optimal policy assigns either 1 or 0 to each state-action pair, which is obtained when optimality is achieved. Please note that an optimal policy is deterministic, i.e., either 1 or 0 for each state-action pair. The authors believe that this fact may be the cause of the reviewer’s confusion. In other words, during the training process, a non-optimal policy is just like in a quantum superposition state; when the training process ends, an optimal policy is “measured” (when the algorithm is converged). Since both the initialization and the training process are random, it is natural to treat the training process as a quantum superposition state; and an optimal policy after convergence (and the Bellman’s optimality equation is satisfied) is just like being “measured” and becomes deterministic.
> > > > >
> > > > > Furthermore, both the Ising model and lowest-energy state are fundamental in physics. The authors are quite impressed by the fact that the Hamiltonian equation (simple and easy-to-implement) can be used as an add-on term to most actor-critic DRL algorithms (note that we tested over 5 algorithms, i.e., DDPG, PPO, SAC, TD3), and such an add-on term effectively addresses practitioners’ major criticism “unstable”. We are confident that this work will be highly recognized by both NeurIPS community members and industrial practitioners.

---

> > > > > > ### Comment · Reviewer_Zi6K · 2022-08-08
> > > > > > **Thanks**
> > > > > >
> > > > > > I appreciate the author's response to my questions. I like the idea that "during the training process, a non-optimal policy is just like in a quantum superposition state".The motivation appears much more clear given the authors' explanations. I am happy to raise my score if the authos promise to complement the manuscript with the following updates:
> > > > > >
> > > > > > 1. Discussion about related work of variance inhibiting in DRL.
> > > > > > 2. Discussion about how this work differs from the paper "K-spin Hamiltonian for quantum-resolvable markov decision processes".
> > > > > > 3. Addressing the potentially confusing points appeared in the reviews from me and others.

---

> > > > > > > ### Author Response · Authors · 2022-08-09
> > > > > > > **Response to Reviewer Zi6K**
> > > > > > >
> > > > > > > Thanks for agreeing to raise the score, and very much appreciate your open attitude toward discussions.
> > > > > > >
> > > > > > > The authors agree that the above three updates (related work of variance inhibiting in DRL, the difference from [23], and clarifying the confusion) are necessary. The authors can promise those updates appearing in the next version.

---

### Author Response · Authors · 2022-08-09
**Summary and Thanks to All Reviewers and Area Chair**

The authors sincerely thank all reviewers and area chair. The authors enjoy the discussions and are happy that some key points reached a consensus.

To recap, this work has made the following major contributions.
1. Per Reviewer 65t2 and Reviewer Reviewer UzCH’s suggestion, the authors added Appx. E to include theoretical analysis of gradient’s variance reduction brought by the proposed H-term.
1. The authors had an extensive discussion with Reviewer 65t2 about the motivation. This work targets a foundational issue that Bellman’s optimality equation has multiple fixed points—failing to return a consistent one. Such an issue of multiple fixed points is quite common in RL practice, which is suspected to be an unavoidable obstacle when addressing practitioners’ major criticism of the highly unstable performance of existing DRL algorithms.
1. The authors had a fruitful interaction with Reviewer JucZ. As a result, Section 3.2 has been substantially revised to investigate the Ising formulation of MDP/RL from a glass of Monte Carlo Gradient Estimator.
1. An interesting point to restate is that under the Ising model (a quite universal model) analogy of MDP/RL, the Hamiltonian equation has a clear physical meaning, namely, it measures the “energy” of a policy. For this physically-inspired H-term, we derived a variant of the policy gradient estimation, as a regulation term (Alg. 1 in Line 15). Such an add-on term turns out to be rather simple to implement and delivers substantial performance improvements.

Even though there are so many great works in the DRL community, the authors are happy to address such a foundational issue and present to the community a simple and effective add-on H-term, inspired by Ising Model and Hamiltonian equation.

Note that the manuscript has been updated in accordance with  the above responses, mainly Section 3.2 and Appx. E.

---

### Meta-Review · Area_Chair_SNrV · 2022-08-26

**Recommendation:** Reject
**Confidence:** Certain

**Metareview:**

The paper proposes to add a regularisation term H to RL algorithms in order to work around issues caused by the multiple fixed points of the Bellman’s optimality equation. The added H term is inspired by quantum field theory, specifically the K-spin Ising model.
All reviewers thought this was an interesting idea, but by the end of the review period, there remained some problems with this paper. Indeed, this paper is not a theory paper, and there is no mathematical proof that the added H term does accomplish the stated goal of variance reduction. This leaves us with empirical evidence. Unfortunately, as was pointed out by reviewers, "Experiment is limited to the 6 MuJoCo tasks", which is not enough to convince that the algorithm should generally work. Finally, many reviewers were confused by the claim that PPO solves the Bellman Optimality Equation. By the end of the review, not all reviewers were convinced this problem had been resolved. This point should be clarified, and it would be better for the paper to go through a new round of reviews before being accepted for publication.

**Award:**

No

---

### Decision · Program_Chairs · 2022-09-14

Reject